# DeNetDM: Debiasing by Network Depth Modulation

**Silpa Vadakkeeveetil Sreelatha**[*1]  **Adarsh Kappiyath**[*1]  **Abhra Chaudhuri**[1,2,3]  **Anjan Dutta**[1]

[1] University of Surrey  [2] University of Exeter  [3] Fujitsu Research of Europe

{s.vadakkeeveetilsreelatha, a.kappiyath, anjan.dutta}@surrey.ac.uk, abhra.chaudhuri@fujitsu.com

## Abstract

Neural networks trained on biased datasets tend to inadvertently learn spurious correlations, hindering generalization. We formally prove that (1) samples that exhibit spurious correlations lie on a lower rank manifold relative to the ones that do not; and (2) the depth of a network acts as an implicit regularizer on the rank of the attribute subspace that is encoded in its representations. Leveraging these insights, we present DeNetDM, a novel debiasing method that uses network depth modulation as a way of developing robustness to spurious correlations. Using a training paradigm derived from Product of Experts, we create both biased and debiased branches with deep and shallow architectures and then distill knowledge to produce the target debiased model. Our method requires no bias annotations or explicit data augmentation while performing on par with approaches that require either or both. We demonstrate that DeNetDM outperforms existing debiasing techniques on both synthetic and real-world datasets by 5%. The project page is available at https://vssilpa.github.io/denetdm/.

## 1  Introduction

Deep neural networks (DNNs) have made remarkable progress across various domains by delivering superior performance on large-scale datasets. However, while the benefits of training DNNs on large-scale datasets are undeniable, these algorithms also tend to inadvertently acquire unwanted biases Shah et al. (2020), hampering their generalization. For instance, a classifier predominantly trained to recognize camels in desert landscapes could encounter difficulties when attempting to identify a camel situated on a road Kim et al. (2021). While a certain degree of bias can enhance model performance, as exemplified by the assumption that cars usually travel on roads Choi et al. (2020), it remains critical to identify and address unwanted biases. Previous methods to address this problem rely on bias annotations as suggested in Majumdar et al. (2021); Kim et al. (2019); Sagawa et al. (2020); Wang et al. (2020), and may involve predefined bias types, such as texture bias mitigation approach in Geirhos et al. (2019). However, acquiring bias labels with human resources is expensive and time-consuming. Recent studies, including Nam et al. (2020) and Lee et al. (2021), have shifted towards debiasing methods without bias labels, with approaches like Nam et al. (2020) emphasizing bias-aligned samples and reweighting bias-conflicting samples, while others like Lee et al. (2021); Kim et al. (2021) introduce augmentation strategies to diversify bias-conflicting points.

We propose DeNetDM (**De**biasing by **Net**work **D**epth **M**odulation), a novel approach to automatically identify and mitigate spurious correlations in image classifiers without relying on explicit data augmentation or reweighting. We start by showing that a sample set that exhibits bias through spurious correlation of attributes lies on a manifold with an effective dimensionality (rank) lower than its bias-free counterpart. We then leverage this finding to formally derive a relationship between the depth of a network and the true rank of the attribute (not sample) subspace that it encodes. We find for a set of attributes that are equally likely to minimize the empirical risk, a deeper network prefers

---

[*]Equal contribution.

38th Conference on Neural Information Processing Systems (NeurIPS 2024).

to retain those with a lower rank, with a higher probability. This implies that the depth of a network acts as an implicit regularizer in the rank space of the attributes. We find that deeper networks tend to generalize based on bias attributes and shallower networks tend to generalize based on core attributes. This finding is in line with a number of works that show that deeper networks tend to learn low rank solutions in general (Roy and Vetterli, 2007; Huh et al., 2023; Wang and Jacot, 2024). Note, however that prior works do not establish the relationship between network depth and the rank of the attribute subspace, a link we establish in our work for the first time, to the best of our knowledge.

Our theoretical claims are confirmed by our preliminary empirical study on linear feature decodability, which quantifies the extent to which specific data attributes can be accurately and reliably extracted from a given dataset or signal. Our study focuses on the feature decodability of bias and core attributes in the neural networks of varying depths, following the approach outlined in Hermann and Lampinen (2020). Our observations in untrained neural networks reveal that the feature decodability tends to diminish as the networks become deeper. We also investigate how attribute decodability varies with Empirical Risk Minimization (ERM) based training on networks of varying depths.

Our hypothesis posits that in a task requiring deep and shallow branches to acquire distinct information, the deep branch consistently prioritizes bias attributes, while the shallow branch favors core attributes. We utilize a technique inspired by the Product of Experts (Hinton, 2002), where one expert is deeper than the other. Empirical analysis shows that the deep branch becomes perfectly biased and the shallow branch becomes relatively debiased by focusing solely on the core attributes by the end of the training. Since the shallow branch may lack the capacity to capture the nuances of the core attributes adequately due to less depth, we propose a strategy where we train a deep debiased model utilizing the information acquired from both deep (perfectly biased) and shallow (weak debiased) network in the previous phase. Our training paradigm efficiently facilitates the learning of core attributes from bias-conflicting data points to the debiased model of any desired architecture.

In summary, we make the following contributions: (1) We theoretically prove that the deep models prefer to learn spurious correlations compared to shallower ones, supported by empirical analysis of the decodability of bias and core attributes across neural networks of varying depths. (2) Building upon the insights from our decodability experiments, we present a novel debiasing approach that involves training both deep and shallow networks to obtain a desired debiased model. (3) We perform extensive experiments and ablation studies on a diverse set of datasets, including synthetic datasets like Colored MNIST and Corrupted CIFAR-10, as well as real-world datasets, Biased FFHQ, BAR and CelebA, demonstrating an approximate 5% improvement over existing methods.

## 2 Related Works

Several works, such as Hermann and Lampinen (2020); Mehrabi et al. (2021), have highlighted neural networks' vulnerability to spurious correlations during empirical risk minimization training. Recently, various debiasing techniques have emerged, which can be categorized as follows.

**Supervision on bias:** A variety of approaches (e.g., Majumdar et al. (2021); Kim et al. (2019); Sagawa et al. (2020); Wang et al. (2020)) assume readily accessible bias labels for bias mitigation. Some approaches assume prior knowledge on specific bias types without using explicit annotations, like texture bias in Wang et al. (2019); Ge et al. (2021); Geirhos et al. (2019). Recent works such as Karimi Mahabadi et al. (2020); Clark et al. (2019) apply the Product of Experts method to mitigate bias in natural language processing, assuming a biased expert's availability. However, obtaining bias labels can be resource-intensive. In contrast, DeNetDM, our proposed method, does not require pre-access to bias labels or types. Instead, it leverages diverse network architecture depths within the Product of Experts framework to implicitly capture relevant bias and core attributes.

**Utilization of pseudo bias-labels:** Recent approaches avoid explicit bias annotations by obtaining pseudo-labels through heuristics to identify biased samples. One heuristic suggests that biases easy to learn are captured early in training, as seen in Nam et al. (2020); Lee et al. (2021); Liu et al. (2023); Kim et al. (2021); Tiwari and Shenoy (2023); Lee et al. (2023). Nam et al. (2020) employ generalized cross-entropy loss to identify and reweight bias-conflicting points. On the other hand, Lee et al. (2021) augment features of bias-conflicting points for debiasing, while Liu et al. (2023) employ logit correction and group mixup techniques to diversify bias-conflicting samples. Other methods like Sohoni et al. (2020) and Seo et al. (2022) acquire pseudo-bias labels through clustering in biased

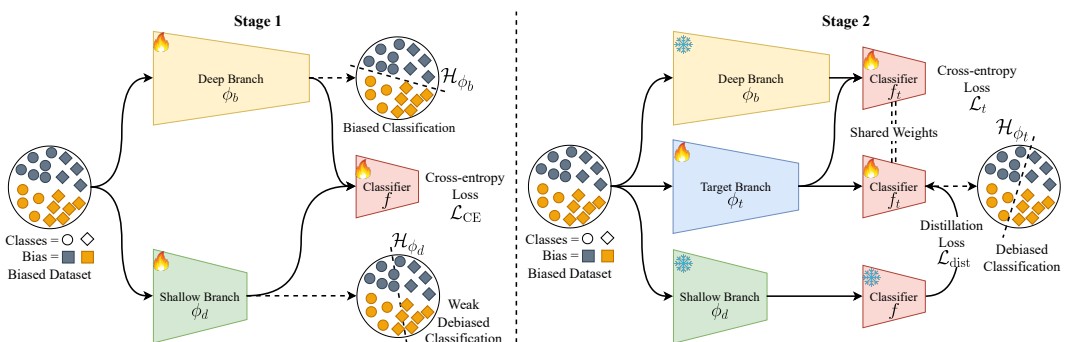

Figure 1: **Illustration of the DeNetDM framework**: In Stage 1, an ensemble of shallow and deep branches produces outputs linearly combined and trained as a product of experts. The cross-entropy loss with depth modulation aids in separating biases and identifying target attributes. In Stage 2, we further introduce a target branch with the desired architecture, which also requires debiasing. This phase exclusively focuses on refining the target branch's feature extractor ($\phi_t$) and classifier head ($f_t$) while leveraging knowledge from the initial stages.

network feature spaces. Our approach does not explicitly require pseudo-bias labels; it implicitly uses them during training to learn both biased and debiased models.

**Dependence on network architectures:** Diffenderfer et al. (2021) employ lottery-ticket-style pruning algorithms for compressed robust architectures. Similarly, approaches like Park et al. (2023); Zhang et al. (2021) introduce pruning to extract robust subnetworks. Our method aligns with this category but does not target specific robust subnetwork discovery. Instead, we utilize training dynamics of varied-depth architectures to enhance debiasing. Meanwhile, Shrestha et al. (2022) applies Occam's razor principle to optimize network depth and visual regions, enhancing overall robustness. Both DeNetDM and OccamNets (Shrestha et al., 2022) aim to simplify learning for better generalization and reduced spurious correlations. DeNetDM uses depth modulation with separate deep and shallow branches to address bias – where the shallow model captures biases and the deep model learns complex, unbiased patterns. In OccamNets, simplicity is a core design principle, with the architecture adaptively minimizing complexity on a per-sample basis. Both methods tackle spurious correlations without extra annotations or data augmentation but through distinct architectural strategies.

## 3 Debiasing by Network Depth Modulation

First, we theoretically justify that the deeper models are more inclined to learn spurious correlations compared to shallow networks, as discussed in Section 3.1. We then provide empirical evidence to support our theoretical claims by utilizing feature decodability, detailed in Section 3.2. Based on these, we introduce DeNetDM, a debiasing approach centered on network depth modulation. Our training process comprises two stages: initially, a deep and shallow network pair is trained using a training paradigm that originates from Products of Experts (Hinton, 2002), yielding both biased and debiased models, which is detailed in Section 3.3. Subsequently, recognizing the limitations of the shallow debiased model in capturing core feature complexities due to its depth, we proceed to train a target debiased model, ensuring it possesses the same or higher depth compared to the deep biased model. This phase leverages information acquired from the biased and debiased models in the previous step, as elaborated in Section 3.4. An illustration of DeNetDM is provided in Figure 1.

**Notations:** We operate on a dataset $X$, where a fraction of the data points, denoted with $X_a$, are bias-aligned and the remaining points, denoted with $X_c$, are bias conflicting. Let $\phi : X \to \mathbb{R}^n$ be an encoder that produces an embedding $z \in \mathbb{R}^n$ for an input $x \in X$. We denote the effective rank (Roy and Vetterli, 2007) of a matrix $A$ as $\rho(A)$, which gives us a continuous notion of the size of the span (rank) of $A$, a quantity that is maximized under equally distributed singular values, and minimized when a single singular value dominates over the rest (Huh et al., 2023). Let $B$ and $C$ be the set of bias and core attributes respectively, both with strictly positive ranks, defining bases that are orthogonal to each other, *i.e.*, $B \perp C$. A summary of notations is provided in Section 7.1.

### 3.1 Simplicity Bias and Spurious Correlations

Debiasing with network depth modulation requires understanding how the depth of a neural network affects its learning of bias-aligned or bias-conflicting subsets of $X$ with lower generalization error. These results finally let us build up to our finding that deeper networks are more susceptible to learning spurious features over their shallower counterparts. All proofs are deferred to Section 7.2.

**Definition 1** (Stability). A partitioning $X = X_1 \cup X_2... \cup X_m$ of a sample set $X$ is stable *wrt.* an attribute $\omega$ when:
$$P(X_i^\omega) = P(X^\omega); \forall i \in [1, m],$$
where $X^\omega$ and $X_i^\omega$ are the respective subspaces of $X$ and $X_i$ corresponding to the attribute $\omega$, and $P(\cdot)$ is the associated probability distribution.

For example, if $\omega$ follows a uniform distribution in $X$, a stable partitioning would ensure that each of the partitions $X_i$ also have $\omega$ distributed uniformly. Stability ensures that a partitioning does not introduce sampling bias into any of the partitions *wrt.* a particular attribute.

**Theorem 1** (Partition Rank). *When the partitioning $X = X_a \cup X_c$ is stable wrt. C, the rank of the bias-aligned partition is upper-bounded by the rank of the bias-conflicting partition, i.e.,*
$$\mathrm{rank}(X_a) \leq \mathrm{rank}(X_c)$$

*Intuition*: The theorem assumes a stable partitioning of the sample set X. It implies that, in both the bias-aligned and conflicting subsets, the distribution of the core attributes are equal to that of the original sample set, *i.e.*, $P(X_a^C) = P(X_c^C) = P(X^C)$. Under this condition, the only component in either of the subsets that determines the subset's rank should be the bias attributes, assuming (without loss of generality) that the attribute space is made up of only the core and the bias attributes. The proof proceeds by establishing that the rank of the bias attributes is lower in the bias-aligned points (resulting from the lack of intra-class variation due to spurious correlation with the class label) than in the bias-conflicting points.

**Theorem 2** (Depth-Rank Duality). *Let $\mathcal{A} = [A_0, A_1, ..., A_n]$ be the attribute subspace of $X$ with increasing ranks, i.e., $\mathrm{rank}(A_0) < \mathrm{rank}(A_1) < ... < \mathrm{rank}(A_n)$, such that every $A \in \mathcal{A}$ is maximally and equally informative of the label $Y$, i.e., $I(A_0, Y) = I(A_1, Y) = ... = I(A_n, Y)$. Then, across the depth of the encoder $\phi$, SGD yields a parameterization that optimizes the following objective:*

$$\underbrace{\min_{\phi, f} \mathcal{L}(f(\phi(X)), Y)}_{ERM} + \min_\phi \sum_d \left\| \phi[d](\tilde{X}) - \Omega^d \odot \mathcal{A} \right\|_2, \tag{1}$$

*where $\mathcal{L}(\cdot, \cdot)$ is the empirical risk, $f(\cdot)$ is a classifier head, $\phi[i](\cdot)$ is the output of the encoder $\phi$ (optimized end-to-end) at depth $d$, $\|\cdot\|_2$ is the $l^2$-norm, $\odot$ is the element-wise product, $\tilde{X}$ is the $l_2$-normalized version of $X$, $\Omega^d = [\mathbb{1}_{\pi_1(d)}; \mathbb{1}_{\pi_2(d)}; ...; \mathbb{1}_{\pi_n(d)}]$, $\mathbb{1}_\pi$ is a random binary function that outputs 1 with a probability $\pi$, and $\pi_i(d)$ is the propagation probability of $A_i$ at depth $d$ bounded as:*

$$\pi_i(d) = \mathcal{O}\left(\rho(\phi[d]) r_i^{-d}\right), \tag{2}$$

*where $\rho(\phi[d])$ is the effective rank of the $\phi[d]$ representation space, and $r_i = \mathrm{rank}(A_i)$.*

*Intuition*: For a set of attributes, all of which equally minimize the training loss, Theorem 2 describes the strategy adopted by SGD to parameterize a neural encoder, for capturing the above set of attributes. At a given depth $d$ of the encoder $\phi$ (represented as $\phi[d]$), each attribute $A_i \in \mathcal{A}$ gets encoded in the representation space of $\phi[d]$ according to its corresponding probability mass $\pi_i(d)$. According to Equation (2), the probability of survival of all attributes decrease with increasing depth. However, the probability of survival of an attribute with a higher rank drops faster with increasing depth than that of one with a lower rank, prioritizing the usage of lower rank attributes at greater depths. In other words, the depth of a network acts as an implicit regularizer in the attribute rank space.

As an example, say, a neural network $\phi$ of depth $d$ (denoted as $\phi[d]$ has $3K$ available dimensions, and of depth $D > d$, $\phi[D]$ has $K$ available dimensions (the rank reduction with increasing depth stemming from the simplicity bias (Huh et al., 2023; Wang and Jacot, 2024)). Say the attribute space it has to learn from is composed of two attributes: (1) $A_0$, with a rank of $K$, and (2) $A_1$, with a rank of $K + i$, where $1 \leq i \leq K$, where both $A_0$ and $A_1$ are equal minimizers of the empirical risk.

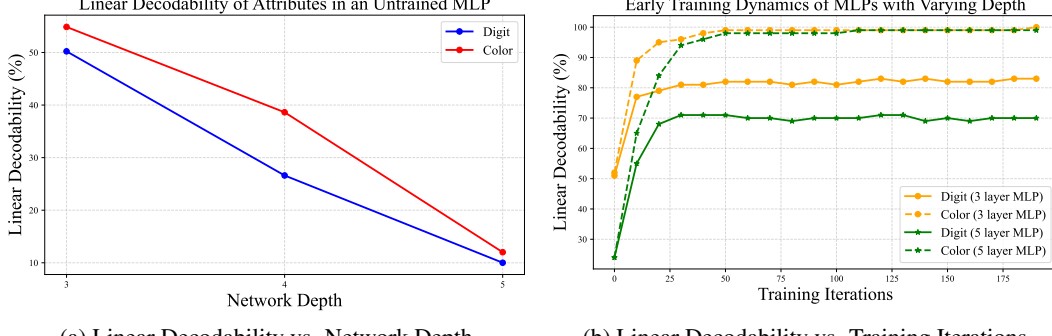

(a) Linear Decodability vs. Network Depth

(b) Linear Decodability vs. Training Iterations

Figure 2: **Exploring the effect of depth modulation:** (a) illustrates how the linear decodability of features decreases as neural network depth increases, while (b) dives into the training dynamics of MLPs with varying depths under ERM.

So, according to Theorem 2, at $\phi[d]$, the encoder has no constraint over the number of attributes it can accommodate, since $3K \geq 2K + i$. However, at depth $D$, $\phi[D]$ can only choose an attribute with $K$ dimensions. Since both $A_0$ and $A_1$ result in the same solution for ERM (Empirical Risk Minimization), SGD would parameterize $\phi[D]$ to capture $A_0$ with a higher probability.

## 3.2 Effect of Depth Modulation

Theorem 2 establishes a relationship between the depth of a network and the nature of the features it learns in terms of its rank. To empirically validate this, we probe MLPs of depths 3, 4, and 5, using the feature decodability technique proposed by Hermann and Lampinen (2020), to uncover the types of features that get encoded in them. We use the Colored MNIST dataset Nam et al. (2020) (CMNIST), where digit identity (core attribute) is spuriously correlated with color (bias attribute). We experiment with the decodability of the digit identity and color attributes in the CMNIST dataset. Additional information on the computation of feature decodability can be found in Section 7.5. We regard digit identity to have a higher rank than that of color, due to its higher representational complexity / information content in terms of the number of bits required for storage, a notion also confirmed in the experiements of Hermann and Lampinen (2020). We start by looking at the decodabilities at random initialization of the networks, and interestingly observe in Figure 2a that the decodabilities of both attributes decrease with increasing depth, but that of digit identity drops faster than color. Since at random initialization, there is no notion of empirical risk, the $\min \mathcal{L}(\cdot, \cdot)$ term in Theorem 2 is cancelled out. Thus, the observation aligns with our prediction of the second term in $\|\cdot\|_2$ of Theorem 2 that the higher the rank of a feature, the less likely it is to get encoded in the later layers, the theoretically predicted behavior specifically for random networks being discussed in Corollary 2.1. We then proceed to investigating how feature decodability evolves during the early stages of Empirical Risk Minimization (ERM) training across the networks of varying depths, *i.e.*, under the presence of $\min \mathcal{L}(\cdot, \cdot)$, the results of which are summarized in Figure 2b. We perform similar linear decodability analysis on C-CIFAR10 dataset and the observations are presented in Section 7.6.1.

As observed in Figure 2b, the initial phases of training for both networks emphasize color attribute (since bias is easy to learn), leading to notable improvements in color decodability for both models. Also, as training progresses, the 3-layer model exhibits higher digit decodability compared to the 5-layer model. Hence, the difference in decodability between color and digit attributes becomes more pronounced in the 5-layer compared to the 3-layer MLP. This again confirms the prediction of our Theorem 2 that when two attributes equally minimize the empirical risk, a deeper network is more likely to select the one with a lower rank, while a shallower network will try to accommodate as much of both as possible. Based on these observations, the deep models may prefer bias attributes, while shallow models focus on core attributes when tasked with capturing distinct information.

This prompts us to explore whether similar behavior can be induced in models of equal depth. In this scenario, both models, undergoing ERM training, may exhibit a similar trend, with the disparity in decodability between biased and core attributes becoming nearly identical in both models due to same depth. Consequently, when compelling each model to learn distinct information, they may

capture biased or core attributes, or even divide attribute information between them, leading to a loss of control over the bias identification process. We also present empirical evidence in Table 5 to support these claims. Therefore, using models of different depths introduces an inductive bias suitable for the bias identification process.

### 3.3 Stage1: Segregation of Bias & Core Attributes

Theorem 1 predicts that bias-aligned points lie on a lower-rank manifold than bias conflicting points. Theorem 2 predicts that as we go deeper into a neural network, the likelihood that a higher rank feature, that equally minimizes the empirical risk as that of other lower rank features, is retained, decays exponentially with depth. Based on this, we present a training procedure to obtain the **b**iased and **d**ebiased classifier for an $M$ class classification problem. Let $\phi_b$ and $\phi_d$ denote the parameters of the feature extractors associated with the deep and shallow branches, where $\mathrm{depth}(\phi_b) > \mathrm{depth}(\phi_d)$. We use $f$ to represent the classifier head shared by $\phi_b$ and $\phi_d$. Here, $f$, $\phi_b$ and $\phi_d$ are trainable parameters. Considering an image-label pair $(x, y)$, the objective function is expressed as:

$$\mathcal{L}_{\mathrm{CE}}(\hat{p}, y) = -\sum_{c=1}^{M} y_c \log(\hat{p}_c) \tag{3}$$

where $\hat{p} = \mathrm{softmax}\left(f(\alpha_b \phi_b(x) + \alpha_d \phi_d(x))\right)$. If we set $\alpha_b = \alpha_d = 1$ throughout the training process, we get:

$$\hat{p} = \mathrm{softmax}\left(f\left(\phi_b(x) + \phi_d(x)\right)\right) \tag{4}$$

To evaluate the performance of an individual expert, we assign a value of 1 to the corresponding $\alpha$ while setting the other $\alpha$ equal to 0.

Our training methodology is derived from the Products of Experts technique (Hinton, 2002) where multiple experts are combined to make a final prediction, and each expert contributes to the prediction with a weight. However, in our approach, the role of the experts is assumed by $\phi_b$ and $\phi_d$, whose features are combined through weighted contributions. The conjunction of features is then passed to the shared classifier to generate predictions. We provide a detailed proof elucidating the derivation of Equation (4) through the Product of Experts in Section 7.3 of the Appendix. Due to the architectural constraints we imposed by modulating their capacities, the deep expert tends to prioritize the learning of bias attribute, while the shallow expert is inclined towards the core attribute. The model leverages the strengths of both experts to effectively learn from their combined knowledge. We investigate the training dynamics in Section 4.3.

### 3.4 Stage2 : Training the Target Debiased Model

The initial phase effectively separates the bias and core attributes into deep and shallow branches, respectively. However, relying solely on the debiased shallow branch may not be practical, as it might not capture the complex features representing the core attributes, given the less depth of the shallow model. This limitation does not apply to the deep biased model. To tackle this challenge, we introduce a target branch with the desired architecture for debiasing.

Let $\phi_t$ be the parameters of the feature extractor associated with the target branch and $f_t$ be the classifier head whose weights are initialized using the weights of $f$. During this phase, our training is exclusively focused on $\phi_t$ and $f_t$. We freeze $\phi_b$ and $\phi_d$ since we leverage these models to only extract the necessary knowledge for debiasing the target branch. To capture information orthogonal to $\phi_b$, we employ the same training approach described in Section 3.3, where $\phi_b$ and $\phi_t$ serve as the experts. The objective function can be written as:

$$\mathcal{L}_t(\hat{p}, y) = -\sum_{c=1}^{M} y_c \log(\hat{p}_c)$$

where $$\hat{p} = \mathrm{softmax}(f_t(\beta_b \phi_b(x) + \beta_t \phi_t(x))) \tag{5}$$

The training and evaluation of the experts follow the procedure described in Section 3.3, with the key difference being that in this phase, only a single expert, $\phi_t$, which is the target branch and classifier $f_t$, undergoes updates.

We further leverage the knowledge pertaining to the core attributes, which is encapsulated in $\phi_d$, by transferring this knowledge to the target branch $\phi_t$ through knowledge distillation. Here, $\phi_t$ acts as the student, whereas $\phi_d$ corresponds to the teacher. We set $\beta_b = 0$ and $\beta_t = 1$ in Equation (5) to obtain the predictions of the student $\phi_t$. Therefore, the distillation loss is given by :

$$\mathcal{L}_{\text{dist}}(\hat{p}_t, \hat{p}_s) = -\sum_{c=1}^{M} \hat{p}_{t_c} \log(\hat{p}_{s_c}) \qquad (6)$$

where $\qquad \hat{p}_s = \text{softmax}\left(\dfrac{f_t(\phi_t(x))}{\tau}\right) \qquad (7) \qquad \hat{p}_t = \text{softmax}\left(\dfrac{f(\phi_d(x))}{\tau}\right) \qquad (8)$

where $\lambda$ is a hyperparameter chosen from the interval $[0, 1]$. The pseudocode for the entire training process of DeNetDM is provided in Section 7.4.

## 4  Experiments

In this section, we discuss the experimental results and analysis to demonstrate the effectiveness of DeNetDM training in debiasing. We evaluate the performance of the proposed approach by comparing it with the previous methods in debiasing, utilizing well-known datasets with diverse bias ratios, consistent with the prior works in debiasing. Additionally, we conduct an empirical study to analyze the training dynamics of DeNetDM. We also perform ablation studies to assess the effectiveness of individual components within the proposed approach.

### 4.1  Experimental Setup

**Datasets:** We evaluate the performance of DeNetDM across diverse domains using two synthetic datasets (Colored MNIST Ahuja et al. (2020), Corrupted CIFAR10 Hendrycks and Dietterich (2019)) and three real-world datasets (Biased FFHQ Kim et al. (2021), BAR Nam et al. (2020)) and CelebA Liu et al. (2015). In Colored MNIST (CMNIST), the digit identity is spuriously correlated with color, while in Corrupted CIFAR10 (C-CIFAR10), the texture noise corrupts the target attribute. Biased FFHQ (BFFHQ) comprises human face images from the FFHQ dataset Karras et al. (2019) such that the age attribute is spuriously correlated with gender. BAR consists of human action images where six human action classes are correlated with six place attributes. We conduct experiments by varying the ratio of bias-conflicting points in the training set to demonstrate the efficacy of our approach across diverse scenarios. Following the experimental settings used by the previous works Liu et al. (2023); Lee et al. (2021); Qi et al. (2022), we vary the ratio of bias-conflicting samples, specifically setting it at {0.5%, 1%, 2%, 5%} for CMNIST and C-CIFAR10, {0.5%} in BFFHQ and {1%, 5%} in BAR datasets. We employ a subsampled version of CelebA as described in Hong and Yang (2021), maintaining the same data splits for consistency.

**Baselines:** We compare the performance of our proposed approach to the following bias mitigation techniques; ERM Vapnik (1999), GDRO Sagawa et al. (2020), LfF Nam et al. (2020), JTT Liu et al. (2021) , DFA Lee et al. (2021) and LC Liu et al. (2023). Among these, GDRO utilizes supervision on bias whereas LfF and JTT assumes no prior knowledge on the bais labels. DFA and LC utilizes augmentation techniques to increase diversity of minority groups. More details on the baselines are provided in Section 7.8.2 of the Appendix.

**Evaluation protocol:** We evaluate CMNIST and C-CIFAR10 on unbiased test sets, with target features randomly correlated to spurious features, following the evaluation protocol commonly used in prior debiasing works Nam et al. (2020); Liu et al. (2021); Lee et al. (2021). Nevertheless, for BFFHQ, we do not use the unbiased test set since half of them are bias-aligned points. To ensure fair evaluation on debiasing, we adhere to previous methods Liu et al. (2023); Lee et al. (2021) by exclusively utilizing a test set comprising bias-conflicting points from the unbiased test set. Notably, the BAR test set consists solely of bias-conflicting samples, posing a significant evaluation challenge. Our primary metric is accuracy, with aligned accuracy and conflicting accuracy calculated separately for some ablations on CMNIST and C-CIFAR10 (see Section 4.4). Aligned accuracy is computed solely on bias-aligned data points while conflicting accuracy is determined exclusively based on the bias-conflicting points. For CelebA, we report worst-group accuracy specifically focusing on the bias-conflicting group (Blonde Hair = 0, Male = 0), which contains a substantial number of samples.

Table 1: Testing accuracy on CMNIST and C-CIFAR10, considering diverse percentages of bias-conflicting samples. Baseline results for C-CIFAR10 are taken from Liu et al. (2023), as we employ the same experimental settings. For CMNIST, we utilize the official repositories to obtain the models. Model requirements for spurious attribute annotations (type) are indicated by ✗ (not required) and ✓ (required).

| Methods | Group | CMNIST | | | | C-CIFAR10 | | | |
|---------|-------|--------|--------|--------|--------|-----------|--------|--------|--------|
| | Info | 0.5 | 1.0 | 2.0 | 5.0 | 0.5 | 1.0 | 2.0 | 5.0 |
| Group DRO | ✓ | 59.67 | 71.33 | 76.30 | 84.40 | 33.44 | 38.30 | 45.81 | 57.32 |
| ERM | ✗ | 35.34 (0.13) | 50.34 (0.16) | 62.29 (1.47) | 77.63 (0.13) | 23.08 (1.25) | 25.82 (0.33) | 30.06 (0.71) | 39.42 (0.64) |
| JTT | ✗ | 53.03 (3.89) | 61.68 (2.02) | 74.23 (3.21) | 85.03 (1.10) | 24.73 (0.60) | 26.90 (0.31) | 33.40 (1.06) | 42.20 (0.31) |
| LfF | ✗ | 63.39 (1.97) | 74.01 (2.21) | 80.48 (0.45) | 85.39 (0.94) | 28.57 (1.30) | 33.07 (0.77) | 39.91 (0.30) | 50.27 (1.56) |
| DFA | ✗ | 59.12 (3.15) | 71.04 (1.02) | 82.86 (2.27) | 88.29 (1.50) | 29.95 (0.71) | 36.49 (1.79) | 41.78 (2.29) | 51.13 (1.28) |
| LC | ✗ | 63.48 (5.22) | 78.41 (1.95) | 83.63 (1.43) | 88.18 (1.59) | 34.56 (0.69) | 37.34 (1.26) | **47.81 (2.00)** | 54.55 (1.26) |
| DeNetDM | ✗ | **74.72 (0.99)** | **85.22 (0.76)** | **89.29 (0.51)** | **93.54 (0.22)** | **38.93 (1.16)** | **44.20 (0.77)** | 47.35 (0.70) | **56.30 (0.42)** |

Table 2: Testing accuracy on BAR, BFFHQ, and CelebA. The test set for BAR and BFFHQ contains only bias-conflicting samples. Baseline method results are derived from Lim et al. (2023) for BAR, Liu et al. (2023) for BFFHQ, and Park et al. (2023) for CelebA on the same dataset split since we utilize identical experimental settings.

| Methods | Group | BAR | | BFFHQ | CelebA |
|---------|-------|-----|-----|-------|--------|
| | Info | 1.0 | 5.0 | 1.0 | - |
| ERM | ✗ | 57.65 (2.36) | 68.60 (2.25) | 56.7 (2.7) | 47.02 |
| JTT | ✗ | 58.17 (3.30) | 68.53 (3.29) | 65.3 (2.5) | 76.80 |
| LfF | ✗ | 57.71 (3.12) | 67.48 (0.46) | 62.2 (1.6) | - |
| DFA | ✗ | 52.31 (1.00) | 63.50 (1.47) | 63.9 (0.3) | 65.26 |
| LC | ✗ | 70.94 (1.46) | 74.32 (2.42) | 70.0 (1.4) | - |
| DeNetDM (ours) | ✗ | **73.84 (2.56)** | **79.61 (3.18)** | **75.7 (2.8)** | **81.04** |

We conduct five independent trials with different random seeds and report both the mean and standard deviation to ensure statistical robustness.

**Implementation details:** We perform extensive hyperparameter tuning using a small unbiased validation set with bias annotations to obtain the deep and shallow branches for all the datasets. We consistently utilize the same debiasing model architectures used by the previous methods for our target branch to ensure a fair comparison. Additionally, a linear layer is employed for the classifier for all the datasets. The additional architecture details for different datasets are as follows: **(1) CMNIST:** we use an MLP with three hidden layers for the deep branch and an MLP with a single hidden layer corresponding to the shallow branch. During the second phase of DeNetDM, we use an MLP with three hidden layers for the target branch. **(2) C-CIFAR10, BAR:** we use the ResNet-20 architecture for the deep branch and a 3-layered CNN model for the shallow branch. The target branch used in the second stage of DeNetDM is ResNet-18. **(3) BFFHQ, CelebA:** we use the ResNet-18 architecture as the biased branch and a 4-layered CNN as the shallow branch. We also use the ResNet-18 architecture for the target branch, following the approaches of Liu et al. (2023); Lee et al. (2021). Further details on the datasets and implementation are presented in Section 7.8.

### 4.2 Evaluation Results

We present a comprehensive comparison of DeNetDM with all the baselines described in Section 4.1 across varying bias conflicting ratios on CMNIST, C-CIFAR10, BFFHQ, BAR and CelebA in Table 1 and Table 2 respectively. As evident from Table 1 and Table 2, DeNetDM consistently outperforms all baselines across different bias ratios for CMNIST, BFFHQ, BAR and CelebA datasets. Notably, on the C-CIFAR10 dataset, DeNetDM exhibits superior performance when bias ratios are at 0.5%, 1%, and 5%, and closely aligns with LC Liu et al. (2023) in the case of 2%. These findings provide evidence for the practical applicability of DeNetDM. It is worth mentioning that the proposed approach demonstrates a significant performance enhancement across all datasets compared to Group DRO, which relies on predefined knowledge of bias. DeNetDM achieves this improvement without any form of supervision on the bias, highlighting the effectiveness of depth modulation in the debiasing.

An intriguing observation from Table 1 is that DeNetDM demonstrates better performance compared to the baselines when the bias-conflicting ratio is lower, particularly evident in the C-CIFAR10 dataset.

We believe that the effectiveness of inductive bias enforced by DeNetDM in distinguishing between core and bias attributes is superior to that of LC, thereby allowing it to adeptly capture core attributes even when dealing with data points that exhibit fewer bias conflicting points. This emphasizes the applicability of DeNetDM in scenarios where the training data exhibits a significant amount of spurious correlations. Another noteworthy observation in Table 2 is that DeNetDM outperforms LC and DFA by a considerable margin across all datasets, particularly on the complex real-world datasets, BAR and BFFHQ. Both LC and DFA rely on augmentations to enhance the diversity of bias-conflicting points, whereas our approach utilizes depth modulation to efficiently capture the core attribute characteristics in the existing training data. Despite this, DeNetDM still achieves superior performance compared to LC and DFA without relying on augmentations.

## 4.3 Analysis of Training Dynamics

In Section 3.2, we discussed the variability in linear decodability at various depths and its significance as a motivation for debiasing. To further validate this intuition and identify the elements contributing to its effectiveness, we delve into the training dynamics of DeNetDM during initial stages. We consider the training of Colored MNIST with 1% skewness due to its simplicity. Figure 3 shows how linear decodability of attributes varies across different branches of DeNetDM during training. As depicted in Figure 3, prior to training, the deep branch demonstrates lower linear decodability for both the digit identity (core attribute) and color (bias attribute) compared to the shallow branch. As training progresses, the bias attribute, easier to learn, rapidly increases in linear decodability in both branches, labeled 'A' in Figure 3.

Here, the disparity in linear decodability between digit identity and color attributes becomes more pronounced in the deep branch than in the shallow one. This distinction serves as a prior, influencing the deep branch to effectively capture the bias. Since we employ Product of Experts technique, the deep branch becomes proficient in classification using the spurious attribute, thereby compelling the shallow branch to rely on other attributes such as digit for the classification. It is worth noting that the linear decodability of core attributes is more pronounced in the shallow branch, allowing them to capture the core attributes. Thus, the training

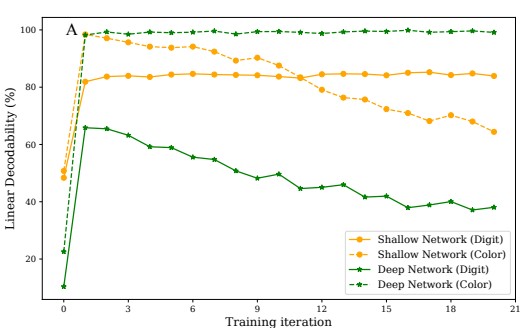

Figure 3: Early training dynamics of DeNetDM.

paradigm of DeNetDM leads to a shallow branch that is robust to spurious correlations, and a deep branch that majorly relies on the biased attribute. This analysis confirms our intuition and provides empirical evidence of effective debiasing.

## 4.4 Ablation Studies

We perform several ablation studies to evaluate different facets of DeNetDM. We scrutinize the effect of various loss components on the performance of DeNetDM. Additionally, we explore the influence of network depth, a fundamental element of DeNetDM, and the sensitivity of DeNetDM to number of parameters which are discussed in Section 7.6. All the experiments are conducted on CMNIST and C-CIFAR10 datasets where the ratio of conflicting points is set to 1%. Additional experiments and ablations are also provided in Section 7.6.

Table 3: Ablation study of different losses used in DeNetDM on C-CIFAR10.

| $\mathcal{L}_{CE}$ (Stage-1) | $\mathcal{L}_{dist}$ (Stage-2) | $\mathcal{L}_t$ (Stage-2) | Accuracy (%) | Conflicting Accuracy (%) | Aligned Accuracy (%) |
|:---:|:---:|:---:|:---:|:---:|:---:|
| ✓ | - | - | 37.47 | 37.42 | 72.40 |
| ✓ | - | ✓ | 42.89 | 35.74 | 81.60 |
| ✓ | ✓ | - | 42.25 | 38.34 | 68.52 |
| ✓ | ✓ | ✓ | 43.12 | 39.46 | 69.53 |

**Effect of loss components:** We conduct ablation studies on C-CIFAR10 by selectively removing components to analyze their impact on the testing set accuracy as well as accuracy on bias-aligned and bias-conflicting points. The results are summarized in Table 3. When considering $\mathcal{L}_{CE}$ alone,

corresponding to the first stage of DeNetDM involving depth modulation, the model achieves 37.42% accuracy, showing a strong ability to learn target attributes. However, introducing the second stage of DeNetDM training with $\mathcal{L}_t$ alone leads to capturing significant bias information alongside core attributes, evidenced by high accuracy on aligned points (81.60%). When introducing $\mathcal{L}_{\text{dist}}$ alone, the model distills knowledge from the shallow branch obtained in the first stage, resulting in performance similar to stage 1 training. However, performing the second stage of DeNetDM training using both $\mathcal{L}_t$ and $\mathcal{L}_{\text{dist}}$ prevents capturing bias, focusing more on learning core features and resulting in improved conflicting and overall accuracy. A similar trend can be observed for CMNIST dataset and the results are summarized in Section 7.7.2.

## 5 Conclusion

We introduce DeNetDM, a novel debiasing method leveraging variations in linear decodability across network depths. Through extensive theoretical and experimental analysis, we uncover insights into the interplay between network architecture, attribute decodability, and training methodologies. DeNetDM employs paired deep and shallow branches inspired by the Product of Experts methodology, transferring debiasing capabilities to the desired architecture. By modulating network depths, it captures core attributes without explicit reweighting or data augmentation. Extensive experiments across various datasets, including synthetic ones like Colored MNIST and Corrupted CIFAR-10, as well as real-world datasets like Biased FFHQ and BAR, validate its robustness and superiority. Importantly, DeNetDM achieves performance comparable to supervised approaches, even without bias annotations.

## 6 Acknowledgments

Silpa Vadakkeveetil Sreelatha is partly supported by the Pioneer Centre for AI, DNRF grant number P1.

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

# 7 Appendix

In the primary text of our submission, we introduce DeNetDM, a novel debiasing framework that leverages the variation of linear decodability across network depths to effectively disentangle bias from core attributes. This technique sets a new benchmark for bias mitigation, achieving unparalleled performance without reliance on data augmentations. To ensure our manuscript's integrity, we provide an extensive appendix designed to complement the main text. This includes a series of additional experiments, expanded ablation studies, comprehensive implementation protocols, and deeper analyses of our findings. The Appendix is presented to bridge the content gap necessitated by the page constraints of the main manuscript, providing a detailed exposition of our methodology and its broader impact on the domain.

## 7.1 Notations

- $B$: Bias attributes
- $C$: Core attributes
- $X, Y$: Sample set (X: Inputs, Y: Labels)
- $X_a$: Bias-aligned points
- $X_b$: Bias-conflicting points
- $\phi$: Encoder
- $\phi[d]$: Encoder at depth $d$
- $\rho(\phi[d])$: Effective rank of an encoder at depth $d$
- $\pi_i(d)$: Propagation probability of an attribute (indexed $i$) at depth $d$
- $\Omega^d$: Propagation probability distribution of an attribute set at layer $d$ of a neural network
- $\mathrm{rank}(\cdot)$: Rank of matrix
- $\dim(\cdot)$: Dimensionality of a tensor / space
- $\varepsilon$: Knock-off probability when transitioning from depth $d$ to $d+1$

## 7.2 Proofs

**Theorem 1** (Partition Rank): When the partitioning $X = X_a \cup X_c$ is stable *wrt.* $C$, the rank of the bias-aligned partition is upper-bounded by the rank of the bias-conflicting partition, *i.e.*,

$$\mathrm{rank}(X_a) \leq \mathrm{rank}(X_c)$$

*Proof.* The theorem assumes a stable partitioning of the sample set X, *i.e.*, in both the bias-aligned and conflicting subsets, the distribution of the core attributes are equal to that of the original sample set, *i.e.*,

$$P(X_a^C) = P(X_c^C) = P(X^C)$$

Under this condition, the only component in either of the subsets that determines the subset's rank should be the bias attributes, under the simplifying assumption (without loss of generality) that the attribute space is made up of only the core and the bias attributes.

For the bias aligned partition $X_a$, all the data points within a class have very low variance within the set of values for the bias attribute $B$, since it is spuriously correlated with the class label. So, $B$ within a class collapses to a much lower dimensional manifold $b \subseteq B$, such that $\mathrm{rank}(b) \leq \dim(B)$. Extending this across classes, without loss of generality, assuming that the number of classes is higher than the variance in $B$ among the bias aligned samples, *i.e.*, $\mathrm{rank}(b)$, $B$ over the set of all classes in $X_a$ would map to a manifold of dimensionality $\mathrm{rank}(b)$. Therefore, since the whole of $B$ in $X_a$ can be represented by a manifold of dimensionality $\mathrm{rank}(b)$ orthogonal to the basis of $C$, the rank of $X_a$ is given by:

$$\mathrm{rank}(X_a) = \mathrm{rank}(C) + \mathrm{rank}(b)$$

For the bias conflicting partition, since there is no correlation between the class labels and $B$, within each class, the bias attributes would require a $\dim(B)$ dimensional subspace independent of $C$, to be represented, since $B \perp C$. This implies that the rank of the bias conflicting points would be:

$$\text{rank}(X_c) = \text{rank}(C) + \dim(B),$$

Since we know that $b \subseteq B$, which leads to $\text{rank}(b) \leq \dim(B)$, it is ultimately implied that $\text{rank}(X_a) \leq \text{rank}(X_c)$.

This completes the proof of the theorem. $\qquad\square$

**Lemma 1.** *Let $\mathcal{A} = [A_0, A_1, ..., A_n]$ be the attribute subspace of $X$ with increasing ranks, i.e., $\text{rank}(A_0) < \text{rank}(A_1) < ... < \text{rank}(A_n)$, such that every $A \in \mathcal{A}$ is maximally and equally informative of the label $Y$, i.e., $I(A_0, Y) = I(A_1, Y) = ... = I(A_n, Y)$. Then, at any given depth $d$ of a neural network, the probability of propagation $\pi_i(d)$ of an attribute $A_i$ is directly proportional to the effective rank $\rho(\phi[d])$ of the network at that depth, i.e.,*

$$\pi_i(d) = \mathcal{O}\left(\rho(\phi[d])\right)$$

*Proof.* Let the total rank of $\mathcal{A}$ be $R$. Consider some reference attribute $A \in \mathcal{A}$ with rank $r$. According to the results on the low rank simplicity bias (Huh et al., 2023; Wang and Jacot, 2024) and deep information propagation (Schoenholz et al., 2017), after propagation through each layer, $\varepsilon R$ of the bases would be knocked off, resulting in a pruned version of $\mathcal{A}$. The total number of ways in which $\mathcal{A}$ can be pruned is given by $\binom{R}{\varepsilon R}$. Also, the number of ways that $A$ features in that pruning is given by $\binom{\varepsilon R}{r}$. Thus, the probability of $A$ being knocked-off in layer-1 of $\phi$ is given by:

$$\binom{\varepsilon R}{r} \Big/ \binom{R}{\varepsilon R} = \frac{r!}{(\varepsilon R - r)! R! (1 - \varepsilon) R!}$$

Therefore, probability of survival at layer $d$:

$$\pi_i(d) = \left( 1 - \frac{r!}{\underbrace{(\varepsilon R - r)!}_{a} R! \underbrace{(1 - \varepsilon) R!}_{b}} \right)^d \tag{9}$$

Therefore, the probability of survival $\pi_i(d)$ of any attribute $A_i$ at depth $d$ increases exponentially with increasing rank $r$ of $A_i$, and decreases exponentially with the knock-off rate $\varepsilon$. $a$ is the part of the knocked-off basis not in $A_i$. $b$ is the part of the complete basis of $\mathcal{A}$ not affected by the first knock-off at layer 1. Thus, at depth $d$, $b^d$ indicates the size of the subspace of $\mathcal{A}$ that survives at depth $d$, therefore being proportional to the effective rank of $\phi[d]$. Based on this, the effective rank at depth $d$ can be written as:

$$\pi_i(d) \propto (1 - \varepsilon)^d R^d = \mathcal{O}((1 - \varepsilon)^d R^d) = \mathcal{O}\left(\rho(\phi[d])\right),$$

This completes the proof of the lemma. $\qquad\square$

**Lemma 2.** *Let $\mathcal{A} = [A_0, A_1, ..., A_n]$ be the attribute subspace of $X$ with increasing ranks, i.e., $\text{rank}(A_0) < \text{rank}(A_1) < ... < \text{rank}(A_n)$, such that every $A \in \mathcal{A}$ is maximally and equally informative of the label $Y$, i.e., $I(A_0, Y) = I(A_1, Y) = ... = I(A_n, Y)$. Then, at any given layer $d$ of a neural network, the propagation probability of an attribute decreases with rank, i.e.,*

$$\pi_1(d) \geq \pi_2(d) \geq ... \geq \pi_n(d),$$

*at a rate that is polynomial in the attribute rank, with degree equal to the depth, i.e.,*

$$\frac{\pi_{i+1}(d)}{\pi_i(d)} = \mathcal{O}(r^{-d})$$

*Proof.* Continuing from Equation (9), we have the propagation probability of $A_i$ at depth $d$ as:

$$\pi_i(d) = \left( 1 - \frac{r!}{\underbrace{(\varepsilon R - r)!R!}_{a} \underbrace{(1-\varepsilon)R!}_{b}} \right)^d$$

Note that when $r$ increases, *i.e.*, for a higher rank attribute, it leads to a drop in $a$, and in a subsequent exponential decrease in $\pi_i(d)$ as follows:

$$\pi_{i+k}(d) = \left( 1 - \frac{(r+k)!}{(\varepsilon R - (r+k))!R!(1-\varepsilon)R!} \right)^d$$
$$\implies \frac{\pi_{i+k}(d)}{\pi_i(d)} \le 1$$
$$\implies \pi_{i+k}(d) \le \pi_i(d)$$
$$\implies \pi_1(d) \ge \pi_2(d) \ge ... \ge \pi_n(d),$$

which proves the first part of the lemma.

Now, taking the ratio of the propagation probabilities of attributes with rank $(i+k)$ and $i$ at depth $d$, we get:

$$\frac{\pi_{i+k}(d)}{\pi_i(d)} = \mathcal{O}\left( \frac{(r+k)^d}{r^d} \right) = \mathcal{O}\left( \left( \frac{r+k}{r} \right)^d \right) = \mathcal{O}\left( \left( 1 + \frac{k}{r} \right)^d \right) = \mathcal{O}(k^d r^{-d})$$

For propagation on to the next layer, $k = 1$. We thus have:

$$\frac{\pi_{i+1}(d)}{\pi_i(d)} = \mathcal{O}(r^{-d})$$

This completes the proof of the lemma. $\qquad\square$

**Theorem 2** (Depth-Rank Duality): Let $\mathcal{A} = [A_0, A_1, ..., A_n]$ be the attribute subspace of $X$ with increasing ranks, *i.e.*, $\text{rank}(A_0) < \text{rank}(A_1) < ... < \text{rank}(A_n)$, such that every $A \in \mathcal{A}$ is maximally and equally informative of the label $Y$, *i.e.*, $I(A_0, Y) = I(A_1, Y) = ... = I(A_n, Y)$. Then, across the depth of the encoder $\phi$, SGD yields a parameterization that optimizes the following objective:

$$\underbrace{\min_{\phi, f} \mathcal{L}(f(\phi(X)), Y)}_{\text{ERM}} + \min_{\phi} \sum_d \left\| \phi[d](\tilde{X}) - \Omega^d \odot \mathcal{A} \right\|_2,$$

where $\mathcal{L}(\cdot, \cdot)$ is the empirical risk, $f(\cdot)$ is a classifier head, $\phi[i](\cdot)$ is the output of the encoder $\phi$ (optimized end-to-end) at depth $d$, $\|\cdot\|_2$ is the $l^2$-norm, $\odot$ is the element-wise product, $\tilde{X}$ is the $l_2$-normalized version of $X$, $\Omega^d = [\mathbb{1}_{\pi_1(d)}; \mathbb{1}_{\pi_2(d)}; ...; \mathbb{1}_{\pi_n(d)}]$, $\mathbb{1}_\pi$ is a random binary function that outputs 1 with a probability $\pi$, and $\pi_i(d)$ is the propagation probability of $A_i$ at depth $d$ bounded as:

$$\pi_i(d) = \mathcal{O}\left( \rho(\phi[d])r_i^{-d} \right),$$

where $\rho(\phi[d])$ is the effective rank of the $\phi[d]$ representation space, and $r_i = \text{rank}(A_i)$.

*Proof.* Since all $A \in \mathcal{A}$ are equally informative about the label Y, they all equally minimize $\mathcal{L}(\cdot, \cdot)$. Thus, the representations learned by $\phi$ are solely determined by the second term in the summation of Equation (1). This means that the SGD must employ a selection mechanism to choose from the $\mathcal{A}$ that optimally utilizes the available parameters in $\mathcal{A}$.

If $\phi[d]$ has sufficiently many parameters to accommodate all of $\mathcal{A}$, SGD should have no reason to discard any of them. However, a number of works that analyze the representational properties of DNNs have found that as we go deeper into a network, the effective number of dimensions available for encoding information, formally known as the effective rank and denoted as $\rho(\phi[d])$ (effective rank of $\phi$ at depth $d$), decreases (Huh et al., 2023; Wang and Jacot, 2024). This characteristic is also known

as the simplicity bias of DNNs. Given the simplicity bias, SGD must learn a parameterization for $\phi$ that optimally selects from $\mathcal{A}$ when the effective rank at a particular layer is lower than $\mathrm{rank}(\mathcal{A})$. In order to stay at the minimum of $\mathcal{L}(\cdot, \cdot)$, $\phi$ must rely on the complete basis of at least one attribute, as only partially learning an attribute would cause deviation from the minimum. So every attribute that is retained for prediction, has to be retained fully. Given this condition, the optimum choice for SGD under constrained effective ranks is thus, to choose $A \in \mathcal{A}$ in increasing order of effective ranks. In other words, the $A_0$ has the highest likelihood of getting chosen, followed by $A_1$, then $A_2$, and so on.

Lemma 1 and Lemma 2 provide bounds for the quantification of the associated probabilities at a given depth, for an attribute of a given rank. Combining them, we get the propagation probability of $A_i$ at depth $d$ as:

$$\pi_i(d) = \mathcal{O}(\rho(\phi[d])r_i^{-d}),$$

where $r_i$ is the rank of $A_i$. We denote the distribution of $\pi$ for an attribute across a network as $\Omega_i$. Without loss of generality, assuming the retention of all attributes at depth $d - 1$, we get the forward pass output at depth $d$ as:

$$\phi[d](X) = \gamma(W_d \cdot \phi[d - 1](X)),$$

where $W_d$ is the weight matrix at layer $d$ and gamma is a non-linearity. Under the most general setting where the elimination of attributes comes only with a decrease in the effective rank and not in the reduction in the dimensionality of the weight matrix, applying Lemmas 1 and 2 we obtain the survival probability of the basis corresponding to all $A \in \mathcal{A}$ in $W$ as:

$$W^d = [\mathbb{1}_{\pi_0(d)} W_0^d; \mathbb{1}_{\pi_1(d)} W_1^d; ...; \mathbb{1}_{\pi_n(d)} W_n^d]$$
$$\implies W^d \cdot X' = [\mathbb{1}_{\pi_0(d)} W_0^d A_0; \mathbb{1}_{\pi_1(d)} W_1^d A_1; ...; \mathbb{1}_{\pi_n(d)} W_n^d A_n]$$
$$= \Omega^d \odot \mathcal{A} \cdot W$$

where $\mathbb{1}_\pi$ is a random binary function that outputs 1 with a probability $\pi$, $\Omega^d = [\mathbb{1}_{\pi_1(d)}; \mathbb{1}_{\pi_2(d)}; ...; \mathbb{1}_{\pi_n(d)}]$, and $X' = \phi[d - 1](X)$. To keep $\mathcal{L}$ at a minimum, $W^d$ must correctly activate for the informative features in $x'$, for which it must maximize $\Omega^d \odot \mathcal{A} \cdot W^d$. Now, $\Omega^d \odot \mathcal{A} \cdot W^d$ is maximized when $W^d = \Omega^d \odot \mathcal{A}$. Thus, the optimal strategy for SGD is to parameterize $W^d$ such that it captures the attributes in $\mathcal{A}$ according to the distribution $\Omega^d$. Over the full depth, the optimization objective would then be:

$$\max_\phi \sum_d \Omega^d \odot \mathcal{A} \cdot \phi[d](X) \equiv \min_\phi \sum_d \left\| \phi[d](\tilde{X}) - \Omega^d \odot \mathcal{A} \right\|_2$$

where $\tilde{X}$ is the $l_2$-normalized version of $X$, and the equivalence comes from the equivalence of maximizing the dot product and minimizing the $l_2$-distance of the normalized samples (Wikipedia, 2024).

This completes the proof of the theorem. $\qquad\square$

**Corollary 2.1.** *Let $\mathcal{A} = [A_0, A_1, ..., A_n]$ be the attribute subspace of $X$ with increasing ranks, i.e., $\mathrm{rank}(A_0) < \mathrm{rank}(A_1) < ... < \mathrm{rank}(A_n)$, such that every $A \in \mathcal{A}$ is maximally and equally informative of the label $Y$, i.e., $I(A_0, Y) = I(A_1, Y) = ... = I(A_n, Y)$. Then, across the depth of a randomly initialized encoder $\phi$, the output of $\phi$ at depth $d$ follows the propagation distribution $\Omega^d$ of the attribute space $\mathcal{A}$ as:*

$$\phi[d](\tilde{X}) \propto \Omega^d \odot \mathcal{A}, \tag{10}$$

*where $\phi[i](\cdot)$ is the output of the encoder $\phi$ at depth $d$, $\odot$ is the element-wise product, $\tilde{X}$ is the $l_2$-normalized version of $X$, $\Omega^d = [\mathbb{1}_{\pi_1(d)}; \mathbb{1}_{\pi_2(d)}; ...; \mathbb{1}_{\pi_n(d)}]$, $\mathbb{1}_\pi$ is a random binary function that outputs 1 with a probability $\pi$, and $\pi_i(d)$ is the probability of propagation of $A_i$ of rank $r_i$ at depth $d$ bounded as:*

$$\pi_i(d) = \mathcal{O}\left(\rho(\phi[d])r_i^{-d}\right), \tag{11}$$

*Discussion*: Let $\mathbb{L}$ be the space of all empirical risks $\{\mathcal{L}_1, \mathcal{L}_2, ...\}$ over $X$. According to the No Free Lunch theorem (Wolpert and Macready, 1997), if an attribute minimizes some $\mathcal{L}_i \in \mathbb{L}$, there exists another $\mathcal{L}_j \in \mathbb{L}$ which it maximizes. So, if we consider the probability of survival of attributes in a randomly initialized network, we need to marginalize the ERM part of Equation (1) across the entirety of $\mathbb{L}$. Assuming an unbiased random initialization scheme, the distribution associated with

$\mathbb{L}$ would be uniform (because no concrete form of empirical risk is defined, we can consider all functions $\mathcal{L} \in \mathbb{L}$ to be equally likely, under the unbiased initialization assumption) as follows:

$$\int_{\mathcal{L} \in \mathbb{L}} \mathcal{L}(f(\phi(X))) \Pr(\mathcal{L}) \, d\mathcal{L},$$

where $\Pr(\mathcal{L})$ is the probability associated with the function $\mathcal{L} \in \mathbb{L}$, which can be assumed to be uniform, as argued before. Then, due to the No Free Lunch Theorem (Wolpert and Macready, 1997), the expected informativeness of all attributes in $X$ is the same, satisfying the $I(A_0, Y) = I(A_1, Y) = ... = I(A_n, Y)$ criterion in the theorem, where the nature of $Y$ is determined by the specific choice of $\mathcal{L}$. The remainder of the reasoning for $\phi[d](\tilde{X}) \propto \Omega^d \odot \mathcal{A}$ is the same as the proof for $\min_\phi \sum_d \left\| \phi[d](\tilde{X}) - \Omega^d \odot \mathcal{A} \right\|_2$ in Theorem 2.

### 7.3 Equivalence with Product of Experts Framework

In Section 3.2 of the main text, we asserted that our training methodology is derived from the Product of Experts. In this section, we elucidate this mathematically:

$$f : \mathbb{R}^F \xrightarrow{\text{linear}} \mathbb{R}^c, \quad \tilde{f}(x) = \text{softmax}(f(x))$$

$$\phi_b : \mathbb{R}^{C \times H \times W} \longrightarrow \mathbb{R}^F, \quad \text{where } F \text{ is the feature dimension}$$

$$\phi_d : \mathbb{R}^{C \times H \times W} \longrightarrow \mathbb{R}^F, \quad \text{such that } \text{depth}(\phi_b) > \text{depth}(\phi_d)$$

$$L(x, y; \phi_b, \phi_d) = -\sum_{c=1}^{C} y_c \log(\hat{p}^c_{\phi_b, \phi_d}) \quad \text{(Loss function definition)}$$

$$\hat{p}_{\phi_b, \phi_d} = \frac{\tilde{f}_c(\phi_b(x)) \cdot \tilde{f}_c(\phi_d(x))}{\sum_{c=1}^{C} \tilde{f}_c(\phi_b(x)) \cdot \tilde{f}_c(\phi_d(x))} \quad \text{(Product of Experts)}$$

$$= \text{softmax}_c(\log(\tilde{f}(\phi_b(x))) + \log(\tilde{f}(\phi_d(x)))) \quad \text{(Softmax log-sum-exp trick)}$$

$$= \text{softmax}_c(f(\phi_b(x)) + f(\phi_d(x))) \quad \text{(Translation invariance of softmax)}$$

$$= \text{softmax}_c(f(\phi_b(x) + \phi_d(x))) \quad \text{(Linearity of classifier } f)$$

We utilize $\hat{p}_{\phi_b, \phi_d}$ to compute the probabilities in DeNetDM which is the same as Equation 2 presented in the main paper.

### 7.4 Pseudocode

The pseudocode for the entire training process of DeNetDM is provided in Algorithm 1.

### 7.5 Feature Decodability

We utilize feature decodability to gauge the extent to which specific dataset features can be reliably decoded across models of varying depths. Hermann and Lampinen (2020) demonstrated that the visual features can be decoded from the higher layers of untrained models. Additionally, they observed that the feature decodability from an untrained model has a significant impact in determining which features are emphasized and suppressed during the model training. Following their approach, we specifically focus on assessing the decodability of bias and core attributes from the penultimate layer of untrained models. In order to evaluate the decodability of an attribute in a dataset, we train a decoder to map the activations from the penultimate layer of a frozen, untrained model to attribute labels. The decoder comprises a single linear layer followed by a softmax activation function. The

**Algorithm 1** DeNetDM: Training

**Input:** Data: $\{(x,y)_i\}_{i=1}^N$
**Output:** $\phi_t, f_t$
**Initialize:** $\phi_t, f_t, f, \phi_b, \phi_d$ such that $\mathrm{depth}(\phi_b) > \mathrm{depth}(\phi_d)$
1: **repeat**
2:     Fetch minibatch data $\{(x,y)_i\}_{i=1}^K$
3:     **for** $i=1$ to $K$ (in parallel) **do**
4:         Compute $\hat{p}$ using (4) to obtain $(\hat{p}, y)_i$
5:     **end for**
6:     Update $\phi_b$, $\phi_d$, $f$ by minimizing $\mathcal{L}_{CE}$ in (3) via SGD
7: **until** Convergence                                   ▷ stage1
8: **repeat**
9:     Fetch minibatch data $\{(x,y)_i\}_{i=1}^K$
10:     **for** $i=1$ to $K$ (in parallel) **do**
11:         Compute $\hat{p}, \hat{p}_s, \hat{p}_t$ via (5), (7), (8) respectively
12:     **end for**
13:     Update $\phi_t, f_t$ by minimizing $\mathcal{L}$ in (6) via SGD
14: **until** Convergence                                  ▷ stage2

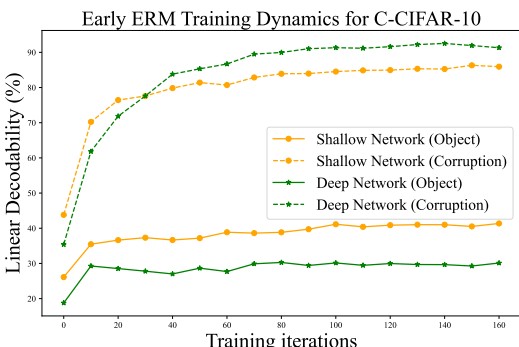

Figure 4: Early training dynamics of DeNetDM on C-CIFAR10 dataset.

decoder is trained using an unbiased validation set associated with the dataset, where each instance is labeled according to the attribute under consideration. Subsequently, the linear decodability of the attribute, measured in accuracy, is reported on the unbiased test set. We investigate the decodability of digit and color attributes in the CMNIST dataset from MLP models with varying depths, including 3, 4, and 5 layers, and the results are depicted in Figure 2a. To investigate how feature decodability evolves during the early stages of Empirical Risk Minimization (ERM) training across networks with varying depths, we train 3-layer and 5-layer MLPs on the CMNIST dataset. Following the training, we evaluate the model's linear decodability for digit and color attributes.

### 7.6 Additional Experiments

#### 7.6.1 Feature Decodability on C-CIFAR10

Analogous to Figure 2 in the main paper, Figure 4 illustrates the variation in feature decodability for corruption (bias) and object (core) in the C-CIFAR10 dataset as ERM training advances. We chose ResNet 20 as the deep network and a 3-layer CNN as the shallow network since these are the architectures used for DeNetDM. The training dynamics show a similar trend to those observed in ColoredMNIST concerning bias and core attributes. As training progresses, corruption (bias) becomes highly decodable by both deep and shallow networks, with the deep branch slightly outperforming the shallow branch. However, the object attribute (core) is more decodable by the shallow network as training progresses, during the initial training dynamics. These observations align with the early training dynamics observed in CMNIST.

## 7.7 Generalization to other tasks

We evaluate the performance of DeNetDM on the CivilComments dataset Koh et al. (2021), which involves natural language debiasing. The task requires classifying online comments as toxic or non-toxic, with labels spuriously correlated with mentions of certain demographic identities. As shown in Table 4, our approach performs comparably to state-of-the-art methods. Due to the constrained rebuttal timeline, we just applied our model out of the box, without any reasonable hyperparameter tuning. The observations illustrate the applicability of DeNetDM to domains beyond vision.

Table 4: Worst group accuracy (%) comparison between different methods on CivilComments dataset.

| Method | Worst Group Acc (%) |
|---|---|
| ERM | 58.6 (1.7) |
| JTT | 69.3 (-) |
| LfF | 58.3 (0.5) |
| LC | 70.3 (1.2) |
| DeNetDM (ours) | 68.33 (-) |

### 7.7.1 Effect of depth modulation

To validate our hypothesis on the significance of network depth in DeNetDM, we conduct an ablation by setting the same depth for both branches and compare it with the default DeNetDM, where one branch is deeper than the other. We focus on the first stage of DeNetDM training for 5 different random seeds, reporting the averaged test accuracy on bias-aligned and bias-conflicting points for individual branches in Table 5. Branch 1 and Branch 2 in Table 5 correspond to the deep and shallow branches in DeNetDM, respectively. We ignore the second stage of training since our focus was primarily on the segregation of bias and core attributes. An interesting observation is the significant standard deviation in accuracies when the branches have the same depth, observed in both datasets. This phenomenon occurs because in such a configuration, DeNetDM loses its ability to clearly distinguish between branches. This is due to the similarity in feature decodability of bias and core attributes across both branches, as discussed in Section 3.2. As a result, DeNetDM may distribute information across multiple branches or still separate core and bias attributes, but the specific branch capturing core attributes varies with different initialization. In contrast, when depths are unequal, the deeper branch tends to focus on aligned points, disregarding conflicting ones, as seen in the test accuracies provided in Table 5. Additionally, the shallow branch emphasizes capturing core attributes, consistently enhancing conflicting accuracy. This shows the pivotal role of depth modulation in the DeNetDM framework for effectively segregating bias and core attributes.

Table 5: Performance of DeNetDM using different network depths for the two branches of DeNetDM.

| Dataset | Depth (Branch 1, Branch 2) | Branch | Conflicting Accuracy (%) | Aligned Accuracy (%) |
|---|---|---|---|---|
| CMNIST | (5, 5) | Branch 1 | **44.94 (22.25)** | 74.85 (12.71) |
| | | Branch 2 | 17.25 (7.89) | **88.57 (9.50)** |
| | (5, 3) | Branch 1 | 1.921 (0.29) | **99.92 (0.25)** |
| | | Branch 2 | **83.17 (0.96)** | 88.25 (2.254) |
| C-CIFAR10 | (ResNet-20, ResNet-20) | Branch 1 | 19.54 (11.16) | 85.83 (8.19) |
| | | Branch 2 | **24.42 (16.93)** | **86.95 (11.04)** |
| | (ResNet-20, 3-layer CNN) | Branch 1 | 3.0 (1.29) | **99.34 (0.47)** |
| | | Branch 2 | **38.52 (0.99)** | 76.72 (2.19) |

### 7.7.2 Effect of loss components on CMNIST

The primary text, constrained by spatial limitations, only includes an ablation study detailing the effect of individual loss components of DeNetDM on the C-CIFAR10 dataset. However, this section extends the scope of our analysis to encompass the CMNIST dataset and the results are summarized

in Table 6. The proposed approach exhibits a similar trend as observed in the case of C-CIFAR10 (presented in Section 4.4).

Table 6: Ablation study of different losses used in DeNetDM on CMNIST dataset.

| $\mathcal{L}_{\text{CE}}$ (Stage-1) | $\mathcal{L}_{\text{dist}}$ (Stage-2) | $\mathcal{L}_t$ (Stage-2) | Accuracy (%) | Conflicting Accuracy (%) | Aligned Accuracy (%) |
|---|---|---|---|---|---|
| ✓ | - | - | 81.61 | 83.28 | 89.66 |
| ✓ | - | ✓ | 82.96 | 81.53 | 95.85 |
| ✓ | ✓ | - | 84.05 | 83.41 | 89.86 |
| ✓ | ✓ | ✓ | 84.97 | 84.44 | 89.17 |

### 7.7.3 Depth vs. Number of parameters

DeNetDM employs depth modulation as its principal strategy for mitigating bias. We investigate the influence of the number of parameters of both branches on DeNetDM performance. We opt for the optimal configuration of the proposed approach on C-CIFAR10 and conducted an ablation study, employing ResNet-20 ($\text{depth}(\phi_b) = 20$) as the deep network and a 3-layer CNN ($\text{depth}(\phi_d) = 3$) as the shallow network. We explore three scenarios where $|\phi_b| < |\phi_d|$, $|\phi_b| \approx |\phi_d|$, and $|\phi_b| > |\phi_d|$. The first stage of DeNetDM training is then performed to analyze learning in the deep and shallow models in each of the cases, and the results are presented in Table 7. As indicated in Table 7, the shallow model exhibits increased resilience to spurious correlations, while the deep model captures bias in all three cases. This suggests that DeNetDM effectively segregates bias and core attributes regardless of the number of parameters in both branches. Interestingly, a notable finding is that the shallow model exhibits better robustness against correlations when the shallow branch possesses a greater number of parameters compared to the deep model, as evident from Table 7.

The findings for CMNIST mirror those observed for C-CIFAR10 as presented in Table 8: the shallow branch demonstrates robustness to spurious correlations, whereas the deep branch consistently assimilates bias irrespective of the number of parameters in both branches. These consistent patterns across datasets reinforce the efficacy of DeNetDM in distinguishing between bias and core attributes.

Table 7: Ablation study on the number of parameters of deep and shallow branches in DeNetDM using C-CIFAR10 dataset.

| Case | Branch | Conflict (%) | Align (%) |
|---|---|---|---|
| $\phi_b > \phi_d$ | $\phi_b$ | 3.08 | **96.8** |
| | $\phi_d$ | **29.78** | 62.61 |
| $\phi_b \approx \phi_d$ | $\phi_b$ | 3.48 | **95.91** |
| | $\phi_d$ | **28.64** | 64.32 |
| $\phi_b < \phi_d$ | $\phi_b$ | 2.04 | **99.01** |
| | $\phi_d$ | **39.05** | 67.68 |

Table 8: Ablation study on the number of parameters of deep and shallow branches in DeNetDM using CMNIST dataset.

| Case | Branch | Conflicting Accuracy (%) | Aligned Accuracy (%) |
|---|---|---|---|
| $\phi_b < \phi_d$ | $\phi_b$ | 11.90 | **99.93** |
| | $\phi_d$ | **83.89** | 88.78 |
| $\phi_b \approx \phi_d$ | $\phi_b$ | 11.87 | **99.90** |
| | $\phi_d$ | **83.07** | 89.09 |
| $\phi_b > \phi_d$ | $\phi_b$ | 10.79 | **98.26** |
| | $\phi_d$ | **83.32** | 88.61 |

### 7.7.4 Effect of Network Depth on DeNetDM

In the main text, we have illustrated how the variation in network depth affects the performance of DeNetDM.We provide an in-depth analysis in this section. As observed in the first three rows of Table 9, as the difference in network depth of deep and shallow progressively increases, the performance of the debiased model increases monotonically. Further, when we decrease the difference in depth of shallow and deep branches (rows 3 and 4) the performance decreases to 80.42% compared to 87.37%. Similar performance degradation can be seen when we increase the depth of the shallow network from 4 to 6 (rows 5 and 6). Hence, DeNetDM is able to distinguish bias and core attributes better when there is a significant difference between the depths of shallow and deep branches. This aligns with the observations presented in Section 3.2 of the main text (Effect of depth modulation).

Table 9: Performance comparison of DeNetDM for various depths of shallow and deep branches.

| Depth (Shallow, Deep) | Conflicting Accuracy (%) | Aligned Accuracy(%) |
|---|---|---|
| (3, 4) | 72.2 | 98.33 |
| (3, 5) | 80.46 | 92.87 |
| (3, 7) | 87.37 | 93.62 |
| (6, 7) | 80.42 | 96.45 |
| (4, 8) | 91.19 | 94.62 |
| (6, 8) | 69.55 | 93.83 |

### 7.7.5 Performance on varying bias-conflicting ratios

We perform experiments on the CMNIST dataset with bias-conflicting ratios of 10% and 20% to evaluate our method's efficacy across a broader range of ratios. The findings, presented in Table 10, show that DeNetDM performs as expected, effectively capturing core attributes in the shallow branch for varied bias ratios.

### 7.7.6 Early training dynamics in ResNet architectures

We also examine the early training dynamics of ResNet-8, ResNet-32, and ResNet-50, akin to Figure 2b in C-CIFAR10 dataset to assess the scalability of DeNetDM to larger ResNet models. After 200 iterations, texture (bias) decodability in all architectures neared 99%, while core attribute decodability for ResNet-8, ResNet-32, and ResNet-50 was 18.74%, 24.32%, and 12.91%, respectively. This aligns with our hypothesis that ResNet-50 would prefer texture attribute over core when paired with ResNet-8 or ResNet-32. To confirm, we tested two setups: (1) ResNet-8 and ResNet-50, and (2) ResNet-32 and ResNet-50. The results, shown in Table 11, indicate high bias-aligned accuracy for ResNet-50 and high bias-conflicting accuracy for ResNet-8 and ResNet-32 respectively. Since ResNet-50 has lower core attribute decodability than ResNet-8 and ResNet-32, it favors bias attributes, while the shallow branches capture core attributes. This experimental results suggest DeNetDM's applicability to diverse, complex and larger models / architectures.

### 7.8 Additional details

In this section, we provide an in-depth discussion of various datasets used along with finer implementation details that enhance the reproducibility of our method.

Table 10: Results on CMNIST with wider bias conflicting ratios.

| Bias ratio | Branch | Conflicting Accuracy (%) | Aligned Accuracy (%) |
|---|---|---|---|
| 10% | Deep | 1.84(0.5) | **99.14(0.2)** |
| | Shallow | **93.12(0.8)** | 96.47(1.3) |
| 20% | Deep | 3.23(2.8) | **97.93(2.1)** |
| | Shallow | **94.49(2.4)** | 97.51(3.4) |

Table 11: Comparison of the performance of DeNetDM using different network depths for the two branches of DeNetDM.

| Depth (Branch 1, Branch 2) | Branch | Conflicting Accuracy (%) | Aligned Accuracy (%) |
|---|---|---|---|
| (ResNet-50, ResNet-32) | Branch 1 | 3.48 (0.98) | **97.15 (2.10)** |
| | Branch 2 | **30.88 (1.22)** | 81.72 (0.73) |
| (ResNet-50, ResNet-8) | Branch 1 | 9.38 (1.52) | **98.60 (0.86)** |
| | Branch 2 | **20.32 (1.90)** | 59.94 (2.61) |

### 7.8.1 Datasets

We provide a detailed description of various datasets used along with a representative sample of all of them.

- **Colored MNIST(CMNIST)**: CMNIST is an adaptation of the MNIST, that introduces color variation to the images. For each digit class, the majority $(1 - \alpha)$ of the images are correlated with the corresponding color $c_i$, with $i$ matching the digit label $y$. The remaining images are randomly assigned one of the other colors $c_j$, where $j \neq y$. The challenge of this dataset lies in identifying the digits despite the strong color bias. To incorporate color variability, a noise vector $v$ drawn from a normal distribution is added to $c_i$. The dataset and its characteristics are illustrated in Figure 5. Among multiple choices of severity, we choose the most severe corruption to simulate the worst-case scenario as done in other works.

- **Corrupted CIFAR10 (C-CIFAR10)**: The Corrupted CIFAR dataset represents an evolved form of the classic CIFAR set, with an emphasis on two particular features: the object depicted and the type of corruption applied. In an approach akin to that used for CMNIST, this dataset adopts an array of corruption styles, labeled from $c_0$, symbolizing blurring, to $c_9$, indicative of snow. Within each object category, a proportion $1 - \alpha$ of the images is intentionally altered with the corruption type $c_i$, corresponding to the object's label $y$. The remainder of the images is processed with a randomly selected corruption type $c_j$, chosen to ensure $j \neq y$. In our dataset, we employ the highest degree of corruption out of the five levels outlined in the original CMNIST dataset. Illustrative samples from this dataset are demonstrated in Figure 5.

- **Biased FFHQ (BFFHQ)**[2] The BFFHQ dataset is a selectively reduced subset derived from the larger FFHQ database of facial images, with a focus on the attributes of gender and age. Gender is designated as the primary attribute of analysis, with age being the secondary attribute that could introduce bias. The gender classification is binary, encompassing male and female categories. The dataset predominantly features male images of subjects aged between 40 and 59, whereas female images are generally of subjects aged between 10 and 29. Samples that defy these age associations—such as younger male or older female subjects—are also present, countering the main age distribution.

- **Biased Action Recognition (BAR)** : The Biased Action Recognition (BAR) dataset comprises real-world images classified into six action categories, each biased towards specific locations. The chosen pairs encompass six common action-location combinations: Climbing on a Rock Wall, Diving underwater, Fishing on a Water Surface, Racing on a Paved Track, Throwing on a Playing Field, and Vaulting into the Sky. The testing set consists solely of samples that present conflicts in bias. Consequently, achieving higher accuracy results on this set indicates superior debiasing performance.

### 7.8.2 Baselines

In this section, we provide a detailed overview of the baselines:

- **Empirical Risk Minimization (ERM)** Vapnik (1999): Standard ERM using cross-entropy loss.

- **Group DRO (GDRO)** Sagawa et al. (2020): A supervised approach that utilizes group labels to identify the worst group and learn an unbiased classifier.

---

[2]`https://github.com/kakaoenterprise/Learning-Debiased-Disentangled`

- **Learning from Failure (LfF)** Nam et al. (2020): Identifies bias-conflicting points through the Generalized Cross Entropy (GCE) loss and upweighting for debiasing.

- **Just Train Twice (JTT)** Liu et al. (2021): Treats misclassified points by ERM-based classifiers as bias-conflicting and upweights them for debiasing.

- **Disentangled Feature Augmentation (DFA)** Lee et al. (2021): Introducing feature augmentation to improve the diversity of bias-conflicting points and enhance unbiased accuracy.

- **Logit Correction (LC)** Liu et al. (2023): Proposes logit correction for bias mitigation along with MixUp Zhang et al. (2018) inspired data augmentation for increasing diversity.

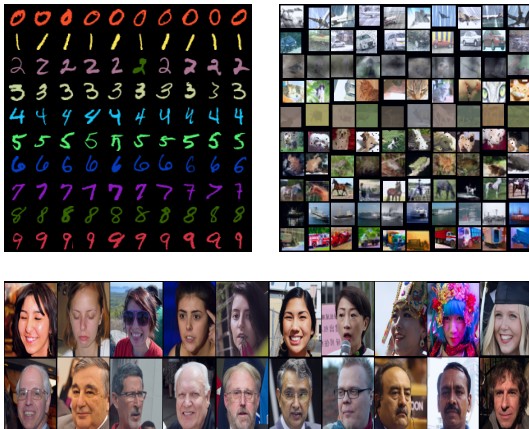

Figure 5: Samples from training data of CMNIST, Corrupted-CIFAR10 and Biased FFHQ.

Table 12: Optimal hyperparameters for the CMNIST, C-CIFAR10, BAR and BFFHQ datasets determined through extensive experimentation. The tuples represent optimal hyperparameters for Stage 1 and Stage 2, respectively.

| Parameter | CMNIST | C-CIFAR10, BAR | BFFHQ |
|---|---|---|---|
| Learning Rate (LR) | $(1.0 \times 10^{-3}, 1.0 \times 10^{-3})$ | $(1.0 \times 10^{-3}, 1.0 \times 10^{-4})$ | $(1.0 \times 10^{-3}, 1.0 \times 10^{-4})$ |
| Batch Size | $(64, 64)$ | $(256, 256)$ | $(64, 64)$ |
| Momentum | $0.9$ | $0.9$ | $0.9$ |
| Weight Decay | $(1.0 \times 10^{-3}, 0)$ | $(1.0 \times 10^{-3}, 0)$ | $(0, 0)$ |
| Epochs | $(100, 100)$ | $(100, 200)$ | $(10, 100)$ |

### 7.8.3 Implementation details

In this section, we detail the optimal hyperparameters identified for various datasets, which were instrumental in achieving the results reported in the main manuscript. The optimal hypeparameters obtained for various datasets are listed in Table 12.Additional parameters not mentioned in Table 12 follow the default values of PyTorch.

**Data Augmentations:** The training phase of DeNetDM incorporated specific data augmentation techniques tailored to each dataset. For instance, the CMNIST dataset did not utilize any form of augmentation. In contrast, the C-CIFAR10 and BFFHQ datasets applied Random Horizontal Flip and random cropping, with the latter involving crops from images padded by 4 pixels. These augmentations are critical as they introduce variability into the dataset, aiding the generalization ability of the neural network.

**Experimental compute:** We utilize RTX 3090 GPUs for all our experiments.

**Architectural Details:** Depth modulation is a critical component of our debiasing strategy. We enumerate the architecture specifics of the shallow branches tailored for each dataset below.

**CMNIST:**

```
(shallow branch): Sequential(
  (c1): Linear(in_features=2352, out_features=100, bias=True)
```

```
    (r1): ReLU()
    (s1): MLPHiddenlayers(
      (hidden_layers): ModuleList(
        (0): Linear(in_features=100, out_features=100, bias=True)
      )
      (act): ReLU()
    )
)
```

## C-CIFAR10 and BAR:

```
(shallow branch): Sequential(
  (c1): Conv2d(3, 32, kernel_size=(5, 5), stride=(1, 1))
  (b1): BatchNorm2d(32, eps=1e-05, momentum=0.1, affine=True,
      track_running_stats=True)
  (r1): ReLU()
  (s1): MaxPool2d(kernel_size=(2, 2), stride=2, padding=0, dilation=1,
      ceil_mode=False)
  (c2): Conv2d(32, 64, kernel_size=(5, 5), stride=(1, 1))
  (b2): BatchNorm2d(64, eps=1e-05, momentum=0.1, affine=True,
      track_running_stats=True)
  (r2): ReLU()
  (s2): MaxPool2d(kernel_size=(2, 2), stride=2, padding=0, dilation=1,
      ceil_mode=False)
  (c3): Conv2d(64, 64, kernel_size=(5, 5), stride=(1, 1))
  (b3): BatchNorm2d(64, eps=1e-05, momentum=0.1, affine=True,
      track_running_stats=True)
  (r3): ReLU()
  (f1): Flatten(start_dim=1, end_dim=-1)
)
(classifier): Linear(in_features=64, out_features=10, bias=True)
(act): ReLU()
```

## BFFHQ:

```
(shallow branch): Sequential(
  (c1): Conv2d(3, 64, kernel_size=(7, 7), stride=(1, 1))
  (b1): BatchNorm2d(64, eps=1e-05, momentum=0.1, affine=True,
      track_running_stats=True)
  (r1): ReLU(inplace=True)
  (s1): MaxPool2d(kernel_size=(2, 2), stride=2, padding=0, dilation=1,
      ceil_mode=False)
  (c2): Conv2d(64, 128, kernel_size=(3, 3), stride=(1, 1))
  (b2): BatchNorm2d(128, eps=1e-05, momentum=0.1, affine=True,
      track_running_stats=True)
  (r2): ReLU(inplace=True)
  (s2): MaxPool2d(kernel_size=(2, 2), stride=2, padding=0, dilation=1,
      ceil_mode=False)
  (c3): Conv2d(128, 512, kernel_size=(3, 3), stride=(1, 1))
  (s3): MaxPool2d(kernel_size=(2, 2), stride=2, padding=0, dilation=1,
      ceil_mode=False)
  (b3): BatchNorm2d(512, eps=1e-05, momentum=0.1, affine=True,
      track_running_stats=True)
  (r3): ReLU(inplace=True)
  (c4): Conv2d(512, 512, kernel_size=(3, 3), stride=(1, 1))
  (b4): BatchNorm2d(512, eps=1e-05, momentum=0.1, affine=True,
      track_running_stats=True)
  (r4): ReLU(inplace=True)
  (a1): AdaptiveAvgPool2d(output_size=(1, 1))
  (f1): Flatten(start_dim=1, end_dim=-1)
)
```

### 7.9 Limitations & Broader Impact

The primary challenge with this approach is the scalability issue when applied to a multi-bias setting. As the number of bias attributes increases, the subtle variations in linear decodability across the various branches could become so refined that accurately identifying biases may fail to achieve high fidelity. Moreover, depending on the network architecture might compel the model to depend excessively on intricate hyperparameter adjustments.

The societal impacts of identifying and mitigating biases in neural networks are extensive, resulting in fairer, more equitable, and trustworthy AI systems. Some of them are as follows :

1. Bias Mitigation in AI : contributes to more equitable AI systems by reducing the influence of spurious correlations.

2. Societal Benefits: contributes to societal fairness by reducing biased decision-making in AI systems and potentially decreases the risk of discrimination in AI applications.

3. Ethical AI Development: encourages transparency and accountability in AI research and deployment.

