# OpenReview forum: "DeNetDM: Debiasing by Network Depth Modulation"
_NeurIPS.cc/2024/Conference — NeurIPS 2024 poster_

### Official Review · Reviewer_Bkbf · 2024-07-12

**Soundness:** 4
**Presentation:** 4
**Contribution:** 3
**Rating:** 6
**Confidence:** 4

**Summary:**

This paper presents a useful theoretical framework that shows that samples that exhibit spurious correlations lie on a lower rank manifold and that the depth of a network acts as an implicit regularizer for the rank of the attribute subspace. Building upon this, the paper proposes a method *DeNetDM* that creates a biased strong encoder (deep network) and a debiased weak encoder (shallow network) and then leverages this to train a final strong encoder that is debiased.

**Strengths:**

1. The theoretical characterization provided in this paper is extremely intuitive and useful. Confirming the intuitions that examples with spurious attributes lie on a lower-dimension manifold formally is the most important contribution of this paper in my opinion. Supplementing this with the explanation that the depth of the network acts as implicit regularizer makes the method of this paper theoretically sound.

2. The confirmation of theoretical findings using synthetic experiments helps validate the theoretical claims.

3. Putting the theoretical findings together into a method that does outperform prior work illustrates the effectivness of this approach.

**Weaknesses:**

1. Empirical Evaluation: The datasets chosen by the authors to evaluate their method are not standard in this literature. Common datasets such as Waterbirds [1] and CelebA [2] are missing. Moreover, including newer more challenging datasets such as UrbanCars [3], SpuCoAnimals [4] and SpuCoMNIST [4] could further improve this paper.

2. Insufficient discussion of similarity to related work: Two related works seem very similar in method to the proposed method. 1) Overcoming simplicity bias in deep networks using a feature sieve and 2) Learning from failure: De-biasing Classifier from biased classifier. The paper would greatly benefit from discussing the similarities and differences between the proposed approach and these 2 closely related approaches. Moreover, Identifying Spurious Biases Early in Training through the Lens of Simplicity Bias is another relevant theoretical / empirical study of spurious correlations which would be useful to compare & contrast with to better understand the contributions of the authors' work.

[1] https://arxiv.org/abs/1911.08731
[2] https://arxiv.org/abs/1411.7766v3
[3] https://arxiv.org/abs/2212.04825
[4] https://arxiv.org/abs/2306.11957

**Questions:**

See above in weaknesses.

**Limitations:**

Empirical evaluation of method: discussed in greater depth in weaknesses.

---

> ### Author Rebuttal · Authors · 2024-08-06
>
> We thank the reviewer for the valuable feedback.
>
> **W1.** Empirical Evaluation: Including newer more challenging datasets could further improve this paper.
>
> **A.** In response to the reviewers' suggestions, we have evaluated the effectiveness of our approach on the CelebA dataset, where blonde hair is the target attribute and gender is the spurious attribute. Due to time constraints, instead of using the full CelebA dataset, we employed a subsampled version as done and described in [A], maintaining the same data splits for consistency. We employed the same architectures that were applied to the BFFHQ dataset, ensuring that the target architecture remains consistent with [A]. To ensure a fair comparison, we reference results from [A] on the same dataset split for methods such as ERM, JTT, Disentangled Feature Augmentation, and DCWP. Currently, we lack results for LC and LfF on this specific dataset version, as they used the original CelebA dataset in their study. We plan to incorporate these comparisons before finalizing the camera-ready version of our paper.
>
> | Method   | Worst Group Accuracy |
> |:--------:|:--------------------:|
> | ERM      |       47.02          |
> | DisEnt   |       65.26          |
> | JTT      |       76.80          |
> | DCWP     |       79.30          |
> | DeNetDM  |       **81.04**          |
>
> As the results indicate, DeNetDM achieves relatively high worst-group accuracy compared to other approaches.
>
> We also conducted experiments on two variants of SpucoMNIST with low and medium spurious difficulty magnitude respectively and compared our results to those reported in [B]. The results were averaged over three random seeds. It can be observed that DeNetDM performs on par with its supervised counterparts, such as GroupDRO and Group Balancing on SpucoMNIST dataset.
>
> | Method           | Worst Group Accuracy (Medium difficulty) | Worst Group Accuracy (Low difficulty) |
> |:----------------:|:----------------------------------------:|:-------------------------------------:|
> | ERM              |              66.0 (5.0)                  |              97.0 (0.5)               |
> | GDRO             |              90.4 (1.9)                  |              96.6 (0.4)               |
> | JTT              |             65.0 (18.4)                  |              94.8 (1.0)               |
> | Group Balancing  |              92.9 (0.1)                  |              95.8 (0.6)               |
> | DeNetDM          |             **93.91 (1.1)**                  |              96.85 (0.1)              |
>
> **W2.** Insufficient discussion of similarity to related work: Two related works seem very similar in method to the proposed method. 1) Overcoming simplicity bias in deep networks using a feature sieve and 2) Learning from failure: De-biasing Classifier from biased classifier.
>
> **A.** We appreciate the reference to the following papers: [C] "Overcoming Simplicity Bias in Deep Networks Using a Feature Sieve," [D] "Learning from Failure: De-biasing Classifiers from Biased Classifiers," and [E] "Identifying Spurious Biases Early in Training through the Lens of Simplicity Bias." A commonality between our work and these approaches is the emphasis on examining the early training dynamics of neural networks to identify and address biases. Specifically, the authors of [C] propose that simple features are learned quickly, appear in early layers of the neural network, and tend to propagate throughout the subsequent layers. Similarly, [D] posits that biases that are easy to learn are captured during the initial phases of training. In [E], the authors demonstrate that, in the early training iterations, the influence of a spurious feature on the network output increases linearly with the level of spurious correlation. While our approach also focuses on the early training dynamics of neural networks, it uniquely characterizes the differences in these dynamics across models with **varying depths** using a Product of Experts architecture, an aspect not explored by the aforementioned approaches. Instead of concentrating on a single neural network, we investigate how the early training dynamics differ between deep and shallow models, demonstrating how core and bias attributes are learned throughout the training process for each model. As suggested by the reviewer, we will incorporate this discussion into the final version of our paper.
>
> [A] Training Debiased Subnetworks with Contrastive Weight Pruning, CVPR 2023.
>
> [B] Towards Mitigating Spurious Correlations in the Wild: A Benchmark & a more Realistic Dataset, Arxiv, Sep 2023.
>
> [C] Overcoming Simplicity Bias in Deep Networks Using a Feature Sieve, ICML 2023.
>
> [D] Learning from Failure: De-biasing Classifiers from Biased Classifiers, NeurIPS 2020.
>
> [E] Identifying Spurious Biases Early in Training through the Lens of Simplicity Bias, AISTATS 2024.

---

> > ### Comment · Reviewer_Bkbf · 2024-08-13
> >
> > I find the author's rebuttal to be convincing and continue to recommend this paper for acceptance.

---

### Official Review · Reviewer_CcWa · 2024-07-12

**Soundness:** 3
**Presentation:** 4
**Contribution:** 3
**Rating:** 6
**Confidence:** 3

**Summary:**

This paper proposes the unsupervised debiasing strategy via modulating network layers depth. It proves that the network with deep layers exploits bias attributes more than that with shallower layers, and shows that training on such less-biased network with shallow layers exhibit debiased learning. The proposed method outperforms other baselines in several datasets including CMNIST, C-CIFAR10, BAR, and BFFHQ.

**Strengths:**

- It is novel contribution that identifies how the rank of attributes, specifically regarding dataset bias, is related to network depth.
- DeNetDM shows superior debiasing performances against existing baselines across different benchmark datasets and correlation severities.

**Weaknesses:**

1. Empirical analysis on networks' depth to learning of different ranks (bias-aligned / bias-conflicting) is limited to CMNIST. Additional experiments on other benchmark datasets, e.g., C-CIFAR10, BAR, and BFFHQ, are required for validity.
2. As shown in Tables 8 and 10, different pairs of networks result in significantly different results (about 20% differences in conflicting accuracy in Table 8). How sensitive are the bias-aligned (bias-conflicting) attributes with regard to depth of deep (shallow) networks, and the following performances of DeNetDM in stage 2?
3. Although authors focus on evaluation on biased dataset, it is unclear whether DeNetDM maintains the accuracy when applied to unbiased dataset.

**Questions:**

See weaknesses.

**Limitations:**

See weaknesses.

---

> ### Author Rebuttal · Authors · 2024-08-06
>
> We thank the reviewer for the valuable feedback and suggestions.
>
> **W1.** Empirical analysis on networks' depth to learning of different ranks (bias-aligned / bias-conflicting) is limited to CMNIST. Additional experiments on other benchmark datasets, e.g., C-CIFAR10, BAR, and BFFHQ, are required for validity.
>
> **A.** Please refer to the global response.
>
> **W2.** As shown in Tables 8 and 10, different pairs of networks result in significantly different results (about 20% differences in conflicting accuracy in Table 8). How sensitive are the bias-aligned (bias-conflicting) attributes with regard to depth of deep (shallow) networks, and the following performances of DeNetDM in stage 2?
>
> **A.** As the reviewer correctly pointed out, Table 8 in the main paper provides insights into how variations in network depth affect the performance of DeNetDM. As shown in Table 8, DeNetDM can better distinguish between bias and core attributes when there is a significant difference between the depths of shallow and deep branches. However, this also depends on the complexity of the features. For instance, in the case of C-CIFAR10, ResNet 32 may capture core features more effectively than ResNet 8 due to the architectural differences. Therefore, pairing ResNet 50 with ResNet 32 in our approach enhances the learning of core attributes in ResNet 32, as reflected in the conflicting accuracy shown in Table 10.
>
> Regarding sensitivity, if the shallow architecture can partially learn the core attributes and there is a significant difference in the depth between the deep and shallow networks, DeNetDM efficiently aids in learning the bias and core attributes within the deep and shallow models, respectively. Debiasing the target architecture in Stage 2 is more effective when the deep and shallow models accurately capture the bias and core attributes respectively in the first stage, allowing us to leverage the distilled information from these networks. Thus, it can be concluded that DeNetDM is sensitive to the depth variations between the deep and shallow branches.
>
> **W3.** It is unclear whether DeNetDM maintains the accuracy when applied to unbiased datasets.
>
> **A.** Our approach operates under the assumption that the data is biased. In such scenarios, deep networks tend to prioritize biased attributes, while shallow models focus on core attributes. When the data is unbiased, the model treats all attributes equally, allowing them to be considered core attributes for the prediction task. In a Product of Experts (PoE) setting, each expert learns a portion of these core attributes, which are then used for joint prediction using both the deep and the shallow branches.
>
> As a result, it cannot be guaranteed that either the deep or shallow branches will perform well individually on the prediction task, since both models contribute to the joint prediction through PoE. We validate this intuition by training stage 1 of DeNetDM on an unbiased CMNIST dataset using a 5-layer MLP and a 3-layer MLP. The deep model and shallow model achieve accuracies of 75.04% and 67.95%, respectively. However, the PoE model, which learns to jointly predict from both deep and shallow experts, achieves an accuracy of 98.64%.
>
> Our approach remains effective if we consider the PoE model from stage 1 as a whole (as the core model) in stage 2 of DeNetDM, rather than focusing on individual shallow or deep models as in the current setting. However, this requires a strategy for model selection (PoE or the shallow model) based on the degree of bias in the data, which could be explored in future work.

---

> > ### Comment · Reviewer_CcWa · 2024-08-11
> >
> > I thank the authors for the rebuttal regarding the concern of architecture sensitivity and unbiased dataset. My further responses are found below:
> >
> > 1. I appreciate the authors' response and the issue has been resolved.
> > 2. I’m still concerned that DeNetDM’s sensitivity requires _significantly large amount of additional hyperparameter tuning_ for network architectures in both shallow and deep networks. Specifically, if there are $N$ and $M$ candidate of architectures for DeNetDM, it requires $N\times M$ validation for choosing the best model, which significantly increases the overall costs for validations when integrated with other hyperparameters. I believe it needs more systematic analysis on how sensitive the DeNetDM is, and therefore how consistently it outperforms other baselines that do not require such tuning, e.g., LfF, DFA.
> > 3. Thanks for the clarification, and I believe such discussion regarding the limitation of DeNetDM when faced with unbiased test set should be included in the paper.

---

> > > ### Author Response · Authors · 2024-08-11
> > > **Response to Reviewer CcWa**
> > >
> > > We thank the reviewer for their valuable feedback. We provide additional clarifications with the hope of addressing the remaining concerns.
> > >
> > > 1,3 : We will include these results in the final revision of the paper.
> > >
> > > 2: We acknowledge the reviewer's point that if there are $N$ and $M$ candidate architectures for DeNetDM, selecting the optimal model could theoretically require up to $N \times M$ validations. However, the list of candidate architectures is manageable because we impose specific constraints on $N$ and $M$ based on our observations and assumptions. Specifically, we assume that the deep branch is either the same depth as the target network or one layer deeper, which limits the hyperparameter search space to $M=2$ across all cases. For the shallow network, we choose $N$ such that the depth of the shallow branch is significantly less than that of the deep branch, as our approach performs better when there is a substantial difference in depth between the shallow and deep networks. This further reduces the search space. For instance, in the case of CMNIST, where the target model is a 5-layer MLP, we consider the following pairs for hyperparameter search: (5,3), (5,4), (6,3), (6,4), and (6,5), which requires only five additional validations. Similarly in the case of C-CIFAR10, where the target model is a ResNet-18, we consider the following pairs for hyperparameter search: (ResNet-18, 3-layer CNN),  (ResNet-18, 4-layer CNN),  (ResNet-18, 5-layer CNN),  (ResNet-20, 3-layer CNN),  (ResNet-20, 4-layer CNN),  (ResNet-20, 5-layer CNN)  which requires only 6 additional validations.
> > > A similar strategy is applied for hyperparameter tuning on other datasets. The best-performing pair from this limited search space is used to report results in Tables 1 and 2. More extensive hyperparameter search might yield better solutions.But, even within the restricted search space, we demonstrate that our approach consistently outperforms all the baselines. Moreover, methods like LfF and DFA also introduce their own set of hyperparameters, such as $q$ in LfF (e.g., $q \in (0.7, 0.8)$) and $(\lambda_{dis}, \lambda_{swapb}, \lambda_{swap})$ in DFA, necessitating method-specific hyperparameter tuning to obtain the best model.
> > >
> > >   The architectures used in Table 10 for C-CIFAR-10 were not originally included in the hyperparameter search space for C-CIFAR-10, as the deep network in question (ResNet-50) is significantly deeper than the target network (ResNet-18). The primary motivation for presenting the experiments in Table 10 is to demonstrate that DeNetDM can scale effectively with more advanced ResNet architectures. As discussed in the initial rebuttal, ResNet32 is more effective in capturing core features than ResNet8, which is a fundamental property of the architecture and not the depth. However, in both the cases, the inductive bias of depth modulation still holds since the deeper branch (ResNet-50) consistently captures bias irrespective of the depth of the shallow network. Hence, the inductive bias relying on depth modulation in DeNetDM is agnostic of such fundamental benefits / weaknesses stemming from  architecture choices, and its sensitivity is a reflection of the best attainable accuracy from any given
> > > architecture.

---

### Official Review · Reviewer_9dFt · 2024-07-12

**Soundness:** 3
**Presentation:** 3
**Contribution:** 2
**Rating:** 5
**Confidence:** 3

**Summary:**

The submission proposes that deeper networks are more likely to use biased features than shallower ones. Using this idea, they develop a training algorithm to encourage reliance upon non-spurious attributes. This is done by training a deep and shallow network as a product-of-experts, then distilling the shallow network into a target network.

Analytic experiments on synthetic datasets indicate that deeper networks do tend to fixate on simpler features as training progresses. A theoretical development of the key intuitions are provided, along with experimental results suggesting that the proposed algorithm can improve over baselines.

**Strengths:**

Originality: The core idea in the submission about depth modulation is original to my knowledge. Some existing work (e.g. Gradient Starvation: A Learning Proclivity in Neural Networks) has discussed how being able to learn some target-correlating features can impede the learning of others, so it seems plausible that a network that learns a simpler feature quicker is less likely to also learn other features.

Quality: There is a good set of comparisons to related methods, on some of the standard datasets in the literature. The ideas presented and the intuitions appear sound. Limitations such as anticipated difficulties in scaling beyond single-biases are acknowledged.

Clarity: The submission is generally intuitively clear and easy to follow.

Significance: The ML community continues to consider ways to approach bias in datasets, since specific domains that are data-scarce and unique enough that foundation models may not be available continue to motivate explorations along these lines.

**Weaknesses:**

One key weakness in approaches of this flavour is the reliance upon use of a “debiased’ validation set to pick crucial hyper-parameters. This presumes knowledge of the bias, and if one knows it exactly, one can likely adopt other bias-informed approaches in practice. There is some work (e.g. Systematic Generalization with Group-Invariant Predictions) that have used a differently-biased validation set to pick hyperparameters relative to the test set, which can be a slightly better alternative than use of an “oracle” validation set.

In practice, it is unclear if a model is always going to be operating in an OOD setting. In fact, it might be the case that OOD settings arise relatively infrequently in deployment. From this perspective, it might make more sense to consider both aligned and conflicting accuracies of all baselines. A competing model that takes less of a hit in-distribution might be preferable.

The learning dynamics of decodability throughout training are interesting, but based only on the simpler datasets with a more drastic difference in the “complexity” of the core and biasing attributes. It is unclear how these trends play out for more realistic data.


Minor:

Is an equation with the full loss for training the target model missing? I’m assuming the objective used for training the target network is really L_t + \lambda*L_dist.

A related baseline might be relevant for inclusion: Simple and Fast Group Robustness by Automatic Feature Reweighting, Qiu et al., ICML 2023.

For the mathematical development, it might be better to include some of the key results from Huh et al., Wang and Jacot, etc. for sake of completeness, and also make it easier to gauge the additional contributions brought in this submission.

Using \citep to might be better to enclose citations in parentheses/brackets.

**Questions:**

The submission mentions use of a “small” validation set, could they clarify the exact size, relative to training and testing (I couldn’t find it easily upon a quick look)?

It would be interesting to look at the decodability dynamics on the more complex datasets such as BFFHQ and BAR. Are there technical difficulties in showcasing these trends that I missed?

Would it be possible to compare both aligned and conflicting accuracies for all methods?

**Limitations:**

The submission acknowledges limitations and broader impact in the Appendix.

---

> ### Author Rebuttal · Authors · 2024-08-06
>
> We appreciate the reviewer's insightful suggestions and comments. Below, we provide detailed responses to each of the questions and concerns raised.
>
> **W1/Q1.** One key weakness in approaches of this flavour is the reliance upon use of a “debiased’ validation set to pick crucial hyper-parameters.
>
> **A.** The authors of [A] provide empirical evidence that group annotations in the validation set are crucial for hyperparameter tuning in bias mitigation. They show that tuning hyperparameters for worst-group validation accuracy significantly improves worst-group test performance compared to tuning for average validation accuracy. Following this approach, as well as the methodology in [B], we tune hyperparameters using a small validation set with group annotations whenever available. For the CMNIST and C-CIFAR10 datasets, we create an unbiased validation set with 250 samples to optimize hyperparameters. In the case of BFFHQ, where an unbiased validation set is not available apart from the test set, we use 40 out of 1000 samples from the test set for validation. For the BAR dataset, we do not perform hyperparameter tuning due to the lack of group annotations and instead apply the same hyperparameters used for the C-CIFAR10 dataset.
>
> We thank the reviewer for citing a reference that employs a differently biased validation set for hyperparameter tuning. However, we believe this is a completely different scenario that warrants independent investigation and could be explored as a future work.
>
> **W2/Q3.** It might make more sense to consider both aligned and conflicting accuracies of all baselines. Would it be possible to compare both aligned and conflicting accuracies for all methods?
>
> **A.** We computed the aligned and conflicting accuracies for all baselines, as well as for our approach, on the BFFHQ dataset. Due to rebuttal time constraints, we could not compare the accuracies with the baselines for the other datasets. As shown in the table, approaches like LfF and DFA reduce accuracy on aligned points while attempting to improve accuracy on conflicting points for BFFHQ dataset. This is likely because these methods focus on enhancing performance in conflicting distribution by reweighing or augmenting minority groups. In contrast, our approach achieves strong performance on conflicting points without sacrificing in-distribution accuracy.
>
> |  Method  | Bias Aligned Acc (%) | Bias Conflicting Acc (%) |
> |:--------:|:---------------------:|:------------------------:|
> |   ERM    |       93.9 (0.2)      |        56.7 (2.7)        |
> |   JTT    |      96.56 (2.9)      |        65.3 (2.5)        |
> |   LfF    |      87.49 (4.5)      |        62.2 (1.6)        |
> |   DFA    |      79.59 (2.7)      |        63.9 (0.3)        |
> |    LC    |      97.8 (1.8)       |        70.0 (1.4)        |
> | DeNetDM  |      **98.55 (0.7)**      |        **75.7 (2.8)**        |
>
>
> **W3/Q2.** The learning dynamics of decodability throughout training are interesting, but based only on the simpler datasets with a more drastic difference in the “complexity” of the core and biasing attributes. It is unclear how these trends play out for more realistic data.
>
> **A**. Please refer to the global response.
>
> **W4.** Minor corrections.
>
> **A.** We thank the reviewer for these minor corrections. We will address them in the final version of the paper.
>
> [A] Liu et al. (2021). Just Train Twice: Improving Group Robustness without Training Group Information, ICML 2021.
>
> [B] Wang et al. (2022). On Feature Learning in the Presence of Spurious Correlations, NeurIPS 2022.

---

> > ### Comment · Reviewer_9dFt · 2024-08-11
> >
> > Thanks for the response! Having read the rebuttal and the other reviews, I am keeping my original rating.

---

### Official Review · Reviewer_2yWj · 2024-07-24

**Soundness:** 3
**Presentation:** 3
**Contribution:** 3
**Rating:** 7
**Confidence:** 4

**Summary:**

This work makes several contributions in relation to network depth and dataset bias. It shows that bias-aligned samples lie on a lower rank manifold compared to bias-conflicting samples. This is linked with network depth by showing how deeper networks tend to prefer spurious correlations, which is demonstrated by the decodability of bias and core features from networks of different depths. Based on these insights, the authors propose DeNetDM, a debiasing approach to train deep (biased) and shallow (debiased) branches, with target debiased model corrected via knowledge distillation. Empirical results demonstrate effectiveness on both synthetic and real-world image classification datasets.

**Strengths:**

[S1] The claims in the paper are backed both theoretically and empirically. Theorem 1 predicts lower-rank manifolds for bias-aligned samples and Theorem 2 shows how deeper networks prefer lower-rank features, thereby providing a solid theoretical backing for the argument that deeper networks prefer bias-aligned samples. The decodability vs depth plots support these claims.

[S2] The proposed debiasing method does not rely on bias annotations or data augmentations, which is advantageous. The efficacy is demonstrated on evaluation benchmarks with varying ratios of biased/unbiased samples, which clearly shows benefits over existing debiasing methods.

**Weaknesses:**

[W1] It is unclear why the biased branch needs to be deep. Clearly, both core and spurious features are available at shallower depths. Would it not be more beneficial to build a shallower bias detector instead? Other works including: [1, 2] have already explored leveraging shallower layers for bias correction, which is more efficient.

[W2] DeNetDM is trained in two stages, but it is unclear why this is necessary. Is it not possible to train both biased branch and perform debiasing in parallel as done in [1]?

[W3] The study does not take into account the impacts of loss functions/regularization. For instance, would the deeper networks exhibit similar proclivity to spurious features with spectral decoupling [3]?

[W4] All the experiments are performed on image classification tasks, which raises questions on generalization to other tasks.

[1] Shrestha, Robik, Kushal Kafle, and Christopher Kanan. "Occamnets: Mitigating dataset bias by favoring simpler hypotheses." European Conference on Computer Vision. Cham: Springer Nature Switzerland, 2022.

[2] Clark, Christopher, Mark Yatskar, and Luke Zettlemoyer. "Learning to model and ignore dataset bias with mixed capacity ensembles." arXiv preprint arXiv:2011.03856 (2020).

[3] Pezeshki, Mohammad, et al. "Gradient starvation: A learning proclivity in neural networks." Advances in Neural Information Processing Systems 34 (2021): 1256-1272.

**Questions:**

The questions correspond to the points listed in the weaknesses section:

[Q1] Why not build a shallower biased branch? Is there any advantage a deeper biased branch offers compared to a shallower one?

[Q2] Does the approach need to be multi-staged? Could the debiasing occur in parallel to the training of the biased branch?

**Limitations:**

The limitations section should mention that is not tested on non-vision/non-classification tasks. Would the approach need any modifications for other tasks?

---

> ### Author Rebuttal · Authors · 2024-08-06
>
> We thank the reviewer for the comments and questions.
>
> **W1/Q1.** Why not build a shallower biased branch?
>
> **A.** TL;DR: Both core and spurious features are indeed available at shallower depths. However, since we do not use explicit bias annotations, we have no a priori way of telling them apart. So, we aim to segregate them automatically. We observe (in Theorem 2) that although both core and spurious features are available at shallower depths, at greater depths, only spurious features are available, while core features are not. So, deep networks provide us with a unique way to identify spurious features (by looking at the features that survive to greater depths), something that is not possible with shallow networks.
>
> Longer answer: In Theorem 2 (via Lemma 2), we show that the propagation probability $\pi_i$ is inversely proportional to the rank of an attribute $r_i$. Specifically, for any two attributes $a_1$ and $a_2$ with ranks $r_1$ and $r_2$ respectively, if $r_2 > r_1$, then at any given depth $d$, $a_1$ is more likely to propagate through depth $d$ than $a_2$. Simply put, attributes with a lower rank would propagate to a greater depth than those with a higher rank.
>
> Since spurious attributes are typically of a much simpler nature (say, color, texture, etc.) relative to core attributes (such as shape), they are more likely to lie on a lower dimensional manifold, effectively having a lower rank. Combining this with Theorem 2, we thus have that for deep networks, spurious attributes survive to greater depths than core attributes. For this reason, deep networks give us a unique way to segregate bias and core attributes that shallow networks cannot offer since both bias and core attributes can survive in the early layers (shallow network) due to their higher propagation probabilities (stemming from higher effective ranks (Huh et al., 2023)), but only spurious / bias attributes can survive up to the later (deeper) layers.
>
> **W2/Q2.** Does the approach need to be multi-staged?
>
> **A.** It is possible to train the deep model, shallow model, and target model in a single stage. In each iteration, the deep and shallow models can be updated using a product of experts (PoE) approach, followed by updating the target model through distillation within the same iteration. We believe that this approach may be less stable than the two-stage method. This instability could arise because the segregation of the deep and shallow models as experts in bias and core attributes, respectively, would not have been established in the early epochs. Consequently, the target model may receive mixed signals during the initial iterations. While parallel training is feasible, it introduces the additional challenge of stabilizing the training dynamics of the target model. Also,   training and backpropagating through all three networks simultaneously demands significant GPU memory resources which can be reduced using multiple stages of training. Due to time constraints, we could not experiment with DeNetDM in this setting, so the single-stage approach could perform better or worse than anticipated.
>
> **W3.** Would deeper networks show similar tendencies toward spurious features with spectral decoupling?
>
> **A.** Thanks for the suggestion, which led to valuable insights. To test the hypothesis that “depth captures biases” under debiasing regularizers like spectral decoupling, we performed ERM on the CMNIST dataset with both deep and shallow networks. We found that while accuracies on bias-aligned points remained near 100%, spectral decoupling(**SD**) improved accuracies on bias-conflicting points more significantly for the deep network than the shallow network (see "Reference to Fig. A in the PDF attached to global response" for results and the best bias-conflicting accuracies in the table below).
>
> |        Method        |  Without SD  |  With SD  | Improvement (With SD - Without SD) |
> |:--------------------:|:------------:|:---------:|:----------------------------------:|
> | Shallow (3-layer) MLP |    43.56     |  47.44    |              +3.88                 |
> | Deep (5-layer) MLP    |    36.90     |  48.74    |             +11.84                |
>
> This indicates that the deep network, being more prone to capturing bias, benefited more from SD in terms of debiasing and accuracy on bias-conflicting points. The shallow network, already focusing on core attributes, saw only marginal improvement. This supports our claim that deeper networks are more bias-prone than shallower ones.
>
> Additionally, the deep network with SD slightly outperformed the shallow network. We hypothesize that SD alleviates rank bottlenecks in deep networks, improving the propagation of core attributes while leveraging the network's larger capacity. This suggests that a deeper network with SD can be both bias-free and more capable, a phenomenon warranting further theoretical exploration.
>
> **W4.** Generalization to other tasks.
>
> **A.** We evaluated the performance of DeNetDM on the CivilComments dataset [A], which involves natural language debiasing. The task requires classifying online comments as toxic or non-toxic, with labels spuriously correlated with mentions of certain demographic identities. As shown in Table 1, our approach performs comparably to state-of-the-art methods. Due to the constrained rebuttal timeline, we just applied our model out of the box, without any reasonable hyperparameter tuning. This was just to illustrate its applicability to domains beyond vision. We believe that with further hyperparameter tuning, we would be able to even surpass the marginal differences that we have with SOTA on CivilComments.
>
> | Method   | Worst Group Acc (%) |
> |:--------:|:-------------------:|
> | ERM      | 58.6 (1.7)          |
> | JTT      | 69.3 (-)            |
> | LfF      | 58.3 (0.5)          |
> | LC       | 70.3 (1.2)          |
> | DeNetDM  | 68.33 (-)           |
>
> [A] WILDS: A benchmark of in-the-wild distribution shifts, ICML 2021.

---

> ### Comment · Reviewer_2yWj · 2024-08-11
>
> I thank the authors for the response. One follow-up I have is:
>
> W1/Q1. Is it necessary to use explicit bias annotations to tell apart the core and spurious features? Existing works [1,4] use bias amplification to segregate the spurious features.
>
> Apart from that, I find the responses for Q2, W3, W4 to be satisfactory. I think the additional results will strengthen the paper. I
>
> [1] Shrestha, Robik, Kushal Kafle, and Christopher Kanan. "Occamnets: Mitigating dataset bias by favoring simpler hypotheses." European Conference on Computer Vision. Cham: Springer Nature Switzerland, 2022.
> [4] Nam, Junhyun, et al. "Learning from failure: De-biasing classifier from biased classifier." Advances in Neural Information Processing Systems 33 (2020): 20673-20684.

---

> ### Author Response · Authors · 2024-08-11
> **Response**
>
> Thank you for your feedback. We are glad that we could address most of your concerns. We will add the additional results to the final revision of the paper.
>
> **W1/Q1**:  As the reviewer rightly pointed out, bias amplification is a well-established method for addressing spurious features without needing explicit bias annotations, as shown in [1], and [4]. However, our approach leverages depth modulation as an alternative strategy for debiasing. Specifically, our deeper branch amplifies bias via depth modulation, a concept grounded in our theoretical insights. Our method also eliminates the need for explicit bias annotations to distinguish between core and spurious features, as the deep and shallow branches inherently act as biased and debiased models, respectively. Furthermore, we demonstrate that DeNetDM outperforms existing approaches like LfF, where bias amplification is achieved using GCE loss.
>
> We hope this clarifies your concern.

---

> > ### Comment · Reviewer_2yWj · 2024-08-11
> >
> > Thank you for the response. I keep my original score of 7, recommending acceptance.

---

### Author Rebuttal · Authors · 2024-08-06

We sincerely thank the reviewers for their valuable feedback and thoughtful comments. We appreciate the opportunity to address the concerns raised and provide clarifications. We offer detailed responses to each reviewer, aiming to address the issues and enhance the clarity of our work. We utilize the global repsonse to address a common concern raised by **Reviewer 9dFt**  and **Reviewer CcWa**.

Reviewer 9dFt raised a concern that the decodability experiments are limited to simple dataset, CMNIST. Reviewer CcWa also raised a concern that the empirical analysis on networks' depth to learning of different ranks (bias-aligned / bias-conflicting) is limited to CMNIST. We provide a common response to both the concerns below:

Analogous to Figure 2b in the main paper, Fig. B in the attached PDF illustrates the variation in feature decodability for corruption (bias) and object (core) in the C-CIFAR10 dataset as ERM training advances. We chose ResNet 20 as the deep network and a 3-layer CNN as the shallow network since these are the architectures used for DeNetDM. The training dynamics show a similar trend to those observed in ColoredMNIST concerning bias and core attributes. As training progresses, corruption  (bias) becomes highly decodable by both deep and shallow networks, with the deep branch slightly outperforming the shallow branch. However, the object attribute (core) is more decodable by the shallow network as training progresses, during the initial training dynamics. These observations align with the early training dynamics observed in CMNIST.

As explained in Section 3.2 of the main text, an unbiased dataset is crucial for training the feature decodable layer to assess attribute decodability. For synthetic datasets like CMNIST and C-CIFAR10, we can generate unbiased datasets. However, this is not the case with natural datasets. The only option is to use existing unbiased test data, but the number of test data points in BFFHQ and BAR is as low as 1,000 and 656, respectively, which we believe is insufficient to train the decodable layer effectively. Additionally, BAR lacks group annotations even for the test set. As a result, we have concerns about the accuracy of feature decodability assessments on these datasets. However, we believe that the early training dynamics observed in C-CIFAR10 provide valuable insights into how these dynamics unfold for complex features.

---

### Decision · Program_Chairs · 2024-09-25

**Decision:**

Accept (poster)

**Comment:**

The paper proposes a novel debiasing method (DeNetDM) that uses network depth modulation to address spurious correlations in biased datasets. The core idea is that deeper networks tend to encode spurious correlations more strongly than shallower networks, which, as reviewer 2yWj pointed out, is an idea that other papers have pursued without requiring bias annotations. The authors present theoretical and empirical evidence to support their approach, demonstrating its effectiveness across various datasets.

**Strengths:**
   - **Theoretical Foundation**: The reviewers appreciated the method's theoretical underpinnings, particularly the proof that bias-aligned samples lie on a lower-rank manifold and how deeper networks are more likely to capture these biases.
   - **Empirical Validation**: The method’s performance on synthetic and real-world datasets is strong.
   - **Novelty**: The approach is novel, although the novelty may be oversold in terms of past work focusing on similar ideas.

**Weaknesses:**
   - **Limited Dataset Evaluation**: Multiple reviewers noted that the empirical analysis was primarily limited to simpler datasets like CMNIST, raising concerns about the method's generalization to more complex, real-world datasets. Many methods fail to generalize from the toy datasets studied here to harder ones (e.g., COCO-on-Places and Biased MNIST).
   - **Sensitivity to Architecture**: There were concerns about the sensitivity of DeNetDM to the choice of network architectures
   - **Lack of Discussion on Related Work**: The paper was critiqued for not adequately discussing related work, especially methods that also address simplicity bias in neural networks. While these papers were cited, e.g., the OccamNet paper, the significant overlap in the core motivation was not discussed. Given that the motivations behind this approach and OccamNets are similar, the paper would be strengthened by a head-to-head comparison, as the baselines are not especially comprehensive.

**Response and Discussion:**
- The authors provided additional experiments.
- The response clarified the rationale for the two-stage training process.
- The authors acknowledged the need for a more thorough discussion of related work and committed to addressing this in the final version of the paper. Given that the authors cited those papers, it is critical that they correctly attribute their ideas.

**Recommendation: Accept.** This paper presents a technically solid contribution to debiasing in machine learning. While there are some concerns regarding the empirical evaluation's scope and the method's sensitivity to architecture, the authors have addressed most of these issues in their rebuttal. However, the authors must address concerns regarding the appropriate attribution of ideas and include an expanded discussion of related work.